# Increased prostaglandin-D₂ in male STAT3-deficient hearts shifts cardiac progenitor cells from endothelial to white adipocyte differentiation

Elisabeth Stelling[1]☯, Melanie Ricke-Hoch[1]☯, Sergej Erschow[1], Steve Hoffmann[2], Anke Katharina Bergmann[3], Maren Heimerl[1], Stefan Pietzsch[1], Karin Battmer[4], Alexandra Haase[5], Britta Stapel[1¤], Michaela Scherr[5], Jean-Luc Balligand[6], Ofer Binah[7], Denise Hilfiker-Kleiner[1]*

1 Department of Cardiology and Angiology, Hannover Medical School, Hannover, Germany,
2 Computational Biology, Leibniz Institute on Aging—Fritz Lipmann Institute (FLI), Jena, Germany,
3 Department of Human Genetics, Hannover Medical School, Hannover, Germany, 4 Department of Hematology, Hemostasis, Oncology and Stem Cell Transplantation, Hannover Medical School, Hannover, Germany, 5 Leibniz Research Laboratories for Biotechnology and Artificial Organs (LEBAO), Department of Cardiothoracic, Transplantation and Vascular Surgery, Hannover Medical School, Hannover, Germany, 6 Pole of Pharmacology and Therapeutics (FATH), Institut de Recherche Expérimentale et Clinique (IREC), Université catholique de Louvain (UCLouvain), Brussels, Belgium, 7 Department of Physiology, Biophysics and Systems Biology, Ruth and Bruce Rappaport Faculty of Medicine and Research Institute, Technion–Israel Institute of Technology, Haifa, Israel

☯ These authors contributed equally to this work.
¤ Current address: Department of Psychiatry, Social Psychiatry and Psychotherapy, Hannover Medical School, Hannover, Germany
* hilfiker.denise@mh-hannover.de

**Data Availability Statement:** All relevant data are within the paper and its Supporting Information files and all files from the genome-wide exploratory

## Abstract

Cardiac levels of the signal transducer and activator of transcription factor-3 (STAT3) decline with age, and male but not female mice with a cardiomyocyte-specific STAT3 deficiency conditional knockout (CKO) display premature age-related heart failure associated with reduced cardiac capillary density. In the present study, isolated male and female CKO-cardiomyocytes exhibit increased prostaglandin (PG)-generating cyclooxygenase-2 (COX-2) expression. The PG-degrading hydroxyprostaglandin-dehydrogenase-15 (HPGD) expression is only reduced in male cardiomyocytes, which is associated with increased prostaglandin D₂ (PGD₂) secretion from isolated male but not female CKO-cardiomyocytes. Reduced HPGD expression in male cardiomyocytes derive from impaired androgen receptor (AR)–signaling due to loss of its cofactor STAT3. Elevated PGD₂ secretion in males is associated with increased white adipocyte accumulation in aged male but not female hearts. Adipocyte differentiation is enhanced in isolated stem cell antigen-1 (SCA-1)⁺ cardiac progenitor cells (CPC) from young male CKO-mice compared with the adipocyte differentiation of male wild-type (WT)-CPC and CPC isolated from female mice. Epigenetic analysis in freshly isolated male CKO-CPC display hypermethylation in pro-angiogenic genes (*Fgfr2*, *Epas1*) and hypomethylation in the white adipocyte differentiation gene *Zfp423* associated with up-regulated ZFP423 expression and a shift from endothelial to white adipocyte differentiation compared with WT-CPC. The expression of the histone-methyltransferase EZH2

methylation profiling by RRBS are available in the Sequencing Read Archive (accession number PRJNA602737).

**Funding:** This work was supported with financial funding by the German Research Foundation (DFG, HI 842/3-2 to D.H.-K.), by the DFG Clinical Research Group (DFG KFO311, HI 842/10-1 to D. H.-K., HI 842/10-2 to D.H.-K., RI 2531/2-1 to M.R.-H., RI 2531/2-2 to M.R.-H.), by the State of Lower Saxony and the Volkswagen Foundation (VWZN3009 to D.H.-K. and M.R.-H.), Hannover, Germany and by REBIRTH I/II as well as the BMBF project de.STAIR (031L0106D to S.H.). The funders had no role in study design, data collection and analysis, decision to publish, or preparation of the manuscript. The authors received no specific funding for this work.

**Competing interests:** The authors have declared that no competing interests exist.

**Abbreviations:** AR, androgen receptor; ARVD, arrhythmogenic right ventricular dysplasia; BAT, brown adipose tissue; BET, beige adipose tissue; BMP2, bone morphogenetic protein 2; BMP4, bone morphogenetic protein 4; cCPC, clonally expanded CPC; CEBPA, CCAAT/enhancer-binding protein alpha; CKO, conditional knockout; CM, cardiomyocyte; COX, cyclooxygenase; CPC, cardiac progenitor cell; DCM, dilated cardiomyopathy; DMEM, Dulbecco's Modified Eagle Medium; DMR, differentially methylated region; DNMT, DNA methyltransferase; DP, $PGD_2$ receptor; EBF2, early B cell factor 2; EGF, epidermal growth factor; EGM-2, endothelial cell growth medium-2; EPO, erythropoietin; EZH2, enhancer of zeste homolog 2; FABP4, fatty acid binding protein 4; FBS, fetal bovine serum; FCS, fetal calf serum; FDR, false discovery rate; FGF, fibroblast growth factor; FITC, fluorescein isothiocyanate; GAPDH, glyceraldehyde 3-phosphate dehydrogenase; HL-1-ctrl-CM, PBS-incubated control HL-1; HL-1-STAT3-KD-CM, HL-1 cardiomyocytes with a lentiviral STAT3-KD; HPGD, hydroxyprostaglandin-dehydrogenase; IB4, isolectin B4; ICM, ischemic cardiomyopathy; IL, interleukin; iPSC, induced pluripotent stem cell; ITS, insulin-transferrin, sodium selenite; I/R, ischemia/reperfusion; KD, knockdown; LV, left ventricular; LVEF, left ventricular ejection fraction; LYZ2, lysozyme 2; MHC, myosin heavy chain; NaCl, sodium chloride; NF, non-failing; PCR2, polycomb repressive complex 2; PDGFRα, platelet-derived growth factor receptor alpha; PG, prostaglandin; $PGD_2$, prostaglandin $D_2$; PGHS-2, prostaglandininsynthase-2; $PGJ_2$, prostaglandin-$J_2$; PPARγ, peroxisome proliferator-activated receptor gamma isoform; PPCM, peripartum cardiomyopathy; PRC2,

is reduced in male CKO-CPC compared with male WT-CPC, whereas no differences in the EZH2 expression in female CPC were observed. Clonally expanded CPC can differentiate into endothelial cells or into adipocytes depending on the differentiation conditions. ZFP423 overexpression is sufficient to induce white adipocyte differentiation of clonal CPC. In isolated WT-CPC, $PGD_2$ stimulation reduces the expression of EZH2, thereby up-regulating ZFP423 expression and promoting white adipocyte differentiation. The treatment of young male CKO mice with the COX inhibitor Ibuprofen or the $PGD_2$ receptor (DP)2 receptor antagonist BAY-u 3405 in vivo increased EZH2 expression and reduced ZFP423 expression and adipocyte differentiation in CKO-CPC. Thus, cardiomyocyte STAT3 deficiency leads to age-related and sex-specific cardiac remodeling and failure in part due to sex-specific alterations in $PGD_2$ secretion and subsequent epigenetic impairment of the differentiation potential of CPC. Causally involved is the impaired AR signaling in absence of STAT3, which reduces the expression of the PG-degrading enzyme HPGD.

## Introduction

Men and women experience quite different cardiovascular disease susceptibility profiles and outcome, a feature that is poorly understood. Further, the effects of biologic sex on health, disease susceptibility, and mortality are vastly understudied [1, 2]. Recent studies showed that genetics contribute to sex-specific differences in fat tissue and cardiovascular and metabolic diseases [3]. Pathophysiologically enhanced cardiac fat content is frequently observed in patients with heart failure, in arrhythmogenic right ventricular dysplasia (ARVD), and after myocardial infarction [4–6]. In patients with dilated cardiomyopathy (DCM), increased fat deposits are associated with more severe left ventricular (LV) dilatation and decreased systolic LV function, compared with DCM patients without enhanced cardiac fat [6]. Three different types of fat tissue exist of which brown adipose tissue (BAT), mostly present in embryonic and fetal stages, and beige adipose tissue (BET) present postnatally, utilize glucose and lipids to generate heat, and are associated with improved cardiometabolic health [7]. Brown and beige adipocytes harbor many similar properties, including multilocular lipid droplets, dense mitochondria, and the activation of a thermogenic gene program involving the PR domain containing 16 (PRDM16) protein. PRDM16 is a transcriptional coregulator that controls the development of brown and beige adipocytes and leads to the up-regulation of uncoupling protein 1 (UCP1), a hallmark of BAT/BET [8, 9]. The third fat type is defined as white adipose tissue (WAT), which stores energy by accumulating fat droplets. WAT is needed as mechanical protection for organs and secretes cytokines and hormones. Extensive WAT formation is associated with an increased cardiovascular risk as it promotes inflammation and alters the immune-endocrine response [10]. WAT-specific gene programs typically up-regulate the zinc finger protein 423 gene (*Zfp423*, the murine ortholog of the human ZNF423). ZFP423 suppresses the PRDM16-mediated thermogenic program, thereby preventing fat cells from burning energy by keeping white fat cells in an energy-storing state [11]. A change in the proportions of adipose tissues occurs with aging, leading to a decline of BAT/BET and an increase in WAT, which plays a central role in cardiovascular diseases and type II diabetes [10, 12]. Development, maintenance, and activation of the different adipose tissues are guided by genetic factors and epigenetic programs, which regulate the de novo differentiation of adipocytes from progenitor cells, as well as white-to-brown adipocyte transdifferentiation [7]). However, little is known about mechanisms driving the buildup of different fat types in the

polycomb repressive complex 2; PRDM16, PR-domain containing 16; PREF-1, preadipocyte factor 1; qRT-PCR, quantitative real-time PCR; rh, recombinant human; rmEPO, recombinant murine erythropoietin; RRBS, reduced representation bisulfite sequencing; SCA-1, stem cell antigen-1; SD, standard deviation; STAT3, signal transducer and activator of transcription factor-3; TMEM26, transmembrane protein 26; UCP-1, uncoupling protein 1; VE, vascular endothelial; WAT, white adipose tissue; WGA, wheat germ agglutinin; WT, wild-type; Zfp423, zinc-finger protein 423; Zfp521, zinc finger protein 521; ZNF, zinc nuclear factor; αMHC, alpha MHC.

heart, especially under pathophysiological conditions and whether there are sex-specific differences in these processes.

The cardiac expression and activation of the signal transducer and activator of transcription factor-3 (STAT3) diminishes with age and is notably reduced in failing hearts from patients with dilatative cardiomyopathy (DCM) or peripartum cardiomyopathy (PPCM) [13–16]. Moreover, cardiomyocyte (CM)-specific deficiency of STAT3 conditional knockout (CKO) leads to age-related heart failure and more pronounced cardiac damage and failure in response to ischemic injury and infection in male mice [17–19]. Nulli-pari CKO females seem to be protected from age-related heart failure but develop PPCM after breeding [14]. Beside direct protective effects on CMs, CM STAT3 influences also the cardiac cell-to-cell communication by regulating the expression and secretion of paracrine factors, impacting on endothelial cells, fibroblasts, inflammatory cells and endogenous stem cell antigen-1 (SCA-1)$^+$ cardiac progenitor cells (CPC) [17, 20–23].

The present study shows that CM-specific STAT3 deficiency leads to an up-regulation of cyclooxygenase-2 (COX-2; also known as prostaglandinsynthase-2, PGHS-2) in young male and female CKO mice, thereby promoting the production of the prostaglandin $D_2$ (PGD$_2$) from arachidonic acid. In males, the prostaglandin degrading enzyme hydroxyprostaglandin-dehydrogenase (HPGD)-15 expression is under the control of the androgen receptor (AR) for which STAT3 acts as a cofactor [24]. As a consequence, PGD$_2$ secretion from male CKO- CM but not from female CKO-CM is increased which subsequently represses the enhancer of zeste homolog 2 (EZH2) subunit of the polycomb repressive complex 2 (PRC2), a histone methyltransferase associated with transcriptional repression of the white adipocyte differentiation factor ZFP423 [25]. In turn, erythropoietin (EPO), which is also reduced in CKO hearts, has been shown to enhance endothelial differentiation from CKO-CPC [22], and the present study shows that it suppresses the ZFP423 expression and white adipocyte differentiation from CKO-CPC.

In summary, STAT3-deficiency leads to sex-specific alterations in the CM secretome, which provokes an epigenetic shift in CPC differentiation from endothelial cells toward white adipocytes in male but not female CKO hearts. This process contributes to a decline in capillaries and an increase in WAT deposits with cardiac remodeling and heart failure in aging male but not female CKO mice.

## Results

### Male but not female CKO mice develop age-related heart failure with increased intraventricular fat accumulation and enhanced inflammation and fibrosis

We previously showed that male and female mice with a CM-specific STAT3 deficiency (alpha MHC [αMHC]-Cre$^{tg/+}$; STAT3$^{flox/flox}$, CKO) exhibit normal cardiac function and morphology at a young age (3 months). Male but not female mice develop left ventricular (LV) systolic dysfunction at 6 months of age (S1 and S2 Tables) [14, 17]. As previously shown in part [14, 17], heart failure in aged male CKO mice is associated with reduced cardiac capillary density (reduced number of capillaries/CM and reduced expression of the endothelial marker vascular endothelial [VE]-cadherin), increased fibrosis (increased collagen deposits and collagen (COL) 1A1 expression) and enhanced inflammation (elevated amount of CD45 positive infiltrates and expression of the macrophage marker epidermal growth factor [EGF]-like module-containing mucin-like hormone receptor-like 1 also known as F4/80 or ADGRE1) compared with hearts from age-matched male wild-type [WT] mice (S1A–S1F Fig). Further histological analyses (Oil Red O and perilipin staining) revealed that LV adipocyte content was low with

no difference between WT and CKO mice of both sexes at the age of 3 to 4 months (male: Fig 1A and 1B, female: S1H Fig). At 6 months, however, LVs from male but not female CKO mice displayed increased adipocytes content compared with age- and sex-matched controls (Fig 1A and 1B, S1H Fig). Further analyses revealed positive staining for the adipocyte marker perilipin and the white adipocyte marker resistin, while the brown/beige adipocyte marker UCP-1 could not be detected (Fig 1C, S1G Fig). In addition, the triglyceride content was higher in LV tissue from 6-month-old male CKO hearts compared with that in age-matched male WT hearts (Fig 1D).

## COX-2 expression is increased in male and female CKO cardiomyocytes but reduced HPGD expression and increased PGD$_2$ secretion are only present in male cardiomyocytes

Molecular analyses revealed a sex-specific difference in the expression of the prostaglandin (PG) degrading enzyme HPGD, with less expression in LV tissue from 3- and 6-month-old CKO male mice compared with that from WT male mice (Fig 1E, S2A Fig), while no such difference was visible between CKO and WT female mice (S2B and S2C Fig). Moreover, HPGD expression was also lower in male CKO-CM compared with that in male WT-CM (Fig 1F), while female CKO-CM and WT-CM showed no difference (S2D Fig). COX-2 plays an important role for the synthesis of PGs. Increased COX-2 expression was observed in LV tissue of 3- and 6-month-old male and female CKO mice and in isolated male and female CMs from young (3-month-old) CKO mice compared with respective sex- and age-matched WT controls (Fig 1G and 1H, S2E–S2H Fig). Furthermore, levels of PGD$_2$ were increased in the supernatants of isolated male CKO-CM but not in female CKO-CM compared with sex-matched WT-CM (Fig 1I and 1J).

## LV tissue from male patients with end-stage heart failure display higher cardiac CEBPA expression and lower HPGD expression compared with non-failing LV samples from healthy male organ donors

We previously showed that STAT3 is reduced in hearts from patients with end-stage heart failure (15). Here, we observed that LV tissue from male patients with end-stage heart failure due to DCM or ischemic cardiomyopathy (ICM) ($n = 8$) displayed higher mRNA levels of the adipocyte marker CCAAT/enhancer binding protein alpha (CEBPA) (+133 ± 142%, $P < 0.05$) and a nonsignificant trend to higher COX-2 (+33 ± 91%, not significant) mRNA levels compared with LV samples from healthy male organ donors ($n = 6$). In addition, HPGD mRNA levels were significantly lower in male failing LV samples (−55 ± 12%, $P < 0.05$) compared with LV samples from healthy male organ donors.

## Serum levels of PGD$_2$ are increased in male patients with heart failure but not in female patients with heart failure

Male patients with heart failure due to idiopathic DCM showed elevated PGD$_2$ serum levels compared with serum from healthy age-matched males, whereas PGD$_2$ serum levels from females with heart failure due to DCM showed no difference compared with serum from healthy age-matched females (Fig 1K and 1L; clinical data of this cohort are shown in S3 Table).

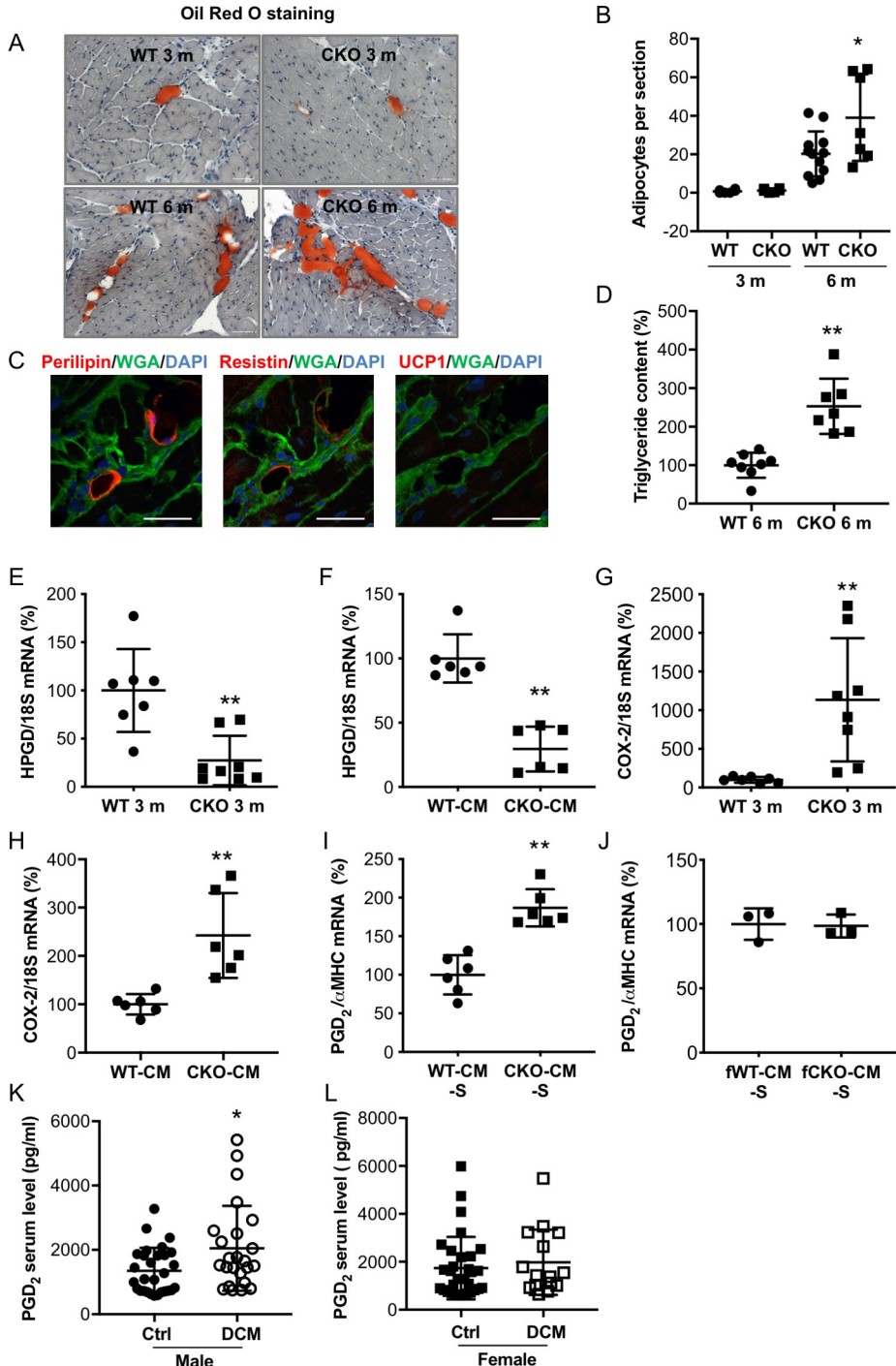

**Fig 1. Male CKO mice with age-related heart failure display increased intraventricular fat accumulation with enhanced inflammation and fibrosis.** (**A**) Oil Red O staining of adipocytes in LV cryosections counterstained with hematoxylin of 3- or 6-month-old (m) male WT or CKO mice, scale bars: 50 μm. (**B**) Bar graph summarizes the number of adipocytes per section in 3- or 6-month-old male WT (3 m: $n = 6$; 6 m: $n = 12$) and CKO (3 m: $n = 6$; 6 m: $n = 7$) LVs, $*$ $p < 0.05$ vs. WT 6 m, 2-way ANOVA with Bonferroni's multiple comparison test. (**C**) Immunofluorescence staining of prilipin (red), resistin (red), or UCP-1 (red) counterstained with WGA-FITC (green) and DAPI (blue) in cryosections of heart tissue (male 6 m CKO mice), scale bars: 25 μM. (**D**) Quantification of triglyceride content in LV extracts from 6 m male WT ($n = 8$) and CKO ($n = 7$) mice, and mean of WT was set at 100%. (**E and F**) Dot plots summarize mRNA levels of HPGD (**E**) in LVs of 3-month-old male WT ($n = 7$) and CKO mice ($n = 8$) and (**F**) in WT-CM and CKO-CM isolated from 3-month old mice ($n = 6$ animals per genotype). (**G and H**) Dot plots summarize COX-2 mRNA levels in (**G**) LVs of 3-month-old male WT ($n = 7$) and CKO mice ($n = 8$) and in (**H**) WT-CM and CKO-CM isolated from 3-month old mice ($n = 6$ animals per genotype). (**I and J**) Measurement of

PGD$_2$ levels (assessed by ELISA, normalized to αMHC mRNA levels) in supernatants of (**I**) male WT-CM and CKO-CM ($n$ = 6 animals per genotype) and in supernatants of (**J**) isolated adult female WT-CM and CKO-CM (CM isolated and pooled from 3 WT and 2 CKO mice). (**K** and **L**) Dot plots depicting PGD$_2$ serum levels (pg/ml) from (**K**) DCM male patients ($N$ = 24) compared with healthy sex-matched controls ($N$ = 29) and from (**L**) DCM females ($N$ = 15) and healthy female controls ($N$ = 31). (**B–J**) All data are mean ± SD, * $p < 0.05$, ** $p < 0.01$ vs. WT, 2-tailed unpaired $t$ test. (**K** and **L**) * $p < 0.05$ vs ctrl, Mann Whitney U test. Underlying data can be found in S1 Data. CKO, conditional knockout; CM, cardiomyocyte; COX, cyclooxygenase; DCM, dilated cardiomyopathy; FITC, fluorescein isothiocyanate; HPGD, hydroxyprostaglandin-dehydrogenase; LV, left ventricular; m, male; PG, prostaglandin; UCP-1, uncoupling protein 1; WGA, wheat germ agglutinin; WT, wild-type; αMHC, alpha MHC.

## HPGD is reduced and COX-2 increased in STAT3-KD HL-1 cardiomyocytes and both effects are enhanced by testosterone but not by estrogen

It has been reported that HPGD is positively regulated by the AR (26) and that the AR is expressed by HL-1 CMs [27]. We observed that HPGD expression was significantly lower in PBS-incubated HL-1 CMs with a lentiviral STAT3-knockdown (KD) (HL-1-STAT3-KD-CM) compared with PBS-incubated control HL-1 (HL-1-ctrl-CM (S3A and S3B Fig). Testosterone stimulation markedly induced HPGD in HL-1-ctrl-CM (+68 ± 16%), which was attenuated in HL-1-STAT3-KD-CM (+29 ± 9%) (S3B Fig). In turn, COX-2 was higher in HL-1-STAT3-KD-CM without an additional effect of testosterone treatment compared with that HL-1-ctrl-CM (S3C Fig). The treatment with estrogen did not influence HPGD or COX-2 expression of HL-1-ctrl-CM or HL-1-STAT3-KD-CM (S3D and S3E Fig).

## CPC isolated from young male CKO mice display increased adipocyte differentiation potential compared with CPC isolated from young male WT mice

Next, we evaluated the differentiation potential of freshly isolated CPC from young WT and CKO male mice for which we previously showed that STAT3 is exclusively deleted in CMs of CKO mice, while its expression is comparable in CPC from CKO (CKO-CPC) and WT mice (22). Freshly isolated CPC showed marked expression of platelet-derived growth factor receptor alpha (PDGFRα) indicative for their mesenchymal stem cell character (Fig 2A and 2B). The lack of preadipocyte factor 1 (PREF-1, highly expressed in the preadipocyte cell line 3T3-L1) expression in CPC confirms that isolated CPC were not contaminated with preadipocytes (Fig 2A and 2B). In vitro cultivation led to spontaneous differentiation into various cell types including endothelial cells and adipocytes, as shown previously [22] (Fig 2C and 2D). The spontaneous adipocyte differentiation was 2-fold higher in male CKO-CPC compared with that in male WT-CPC, despite identical cultivation conditions (Fig 2D and 2E), while no elevated adipocyte differentiation was observed in female WT-CPC and CKO-CPC. After differentiation, male CKO-CPC displayed an up-regulation of the general adipocyte markers CEBPA and fatty acid binding protein 4 (FABP4) (Fig 2F and 2G), and WAT markers ZFP423, lysozyme 2 (LYZ2), and resistin (Fig 2H–2J). The expression of the BAT/BET markers EBF transcription factor-2 (EBF2) and transmembrane protein 26 (TMEM26) was similar in CKO-CPC and WT-CPC (Fig 2K and 2L) and the BAT/BET markers PRDM16 and UCP-1 could not be detected.

## The epigenetic signature of freshly isolated CPC isolated from young CKO males differs from CPC isolated from young WT males

Since adipocyte priming of CKO-CPC was maintained during in vitro cultivation, we analyzed the epigenetic profiles of CKO-CPC and WT-CPC freshly isolated from 3-month-old male

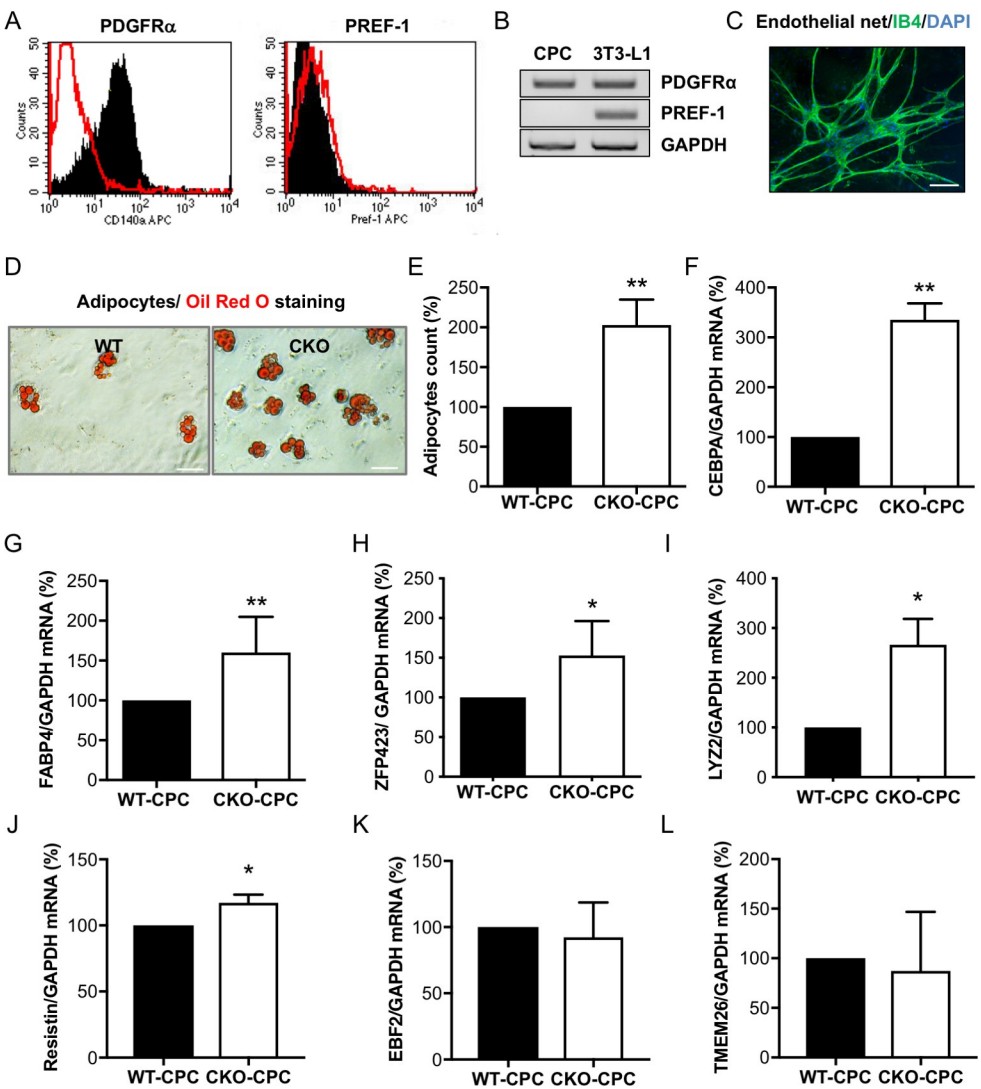

**Fig 2. Characterization and adipocyte formation of cultivated CPC isolated from young CKO and WT male mice with normal cardiac function.** (A) Flow cytometry (PDGFRα or PREF-1: black; IgG control: red) and (B) qRT-PCR analysis of PDGFRα and PREF-1 in freshly isolated male WT-CPC, $n = 3$ independent isolations (each isolation consists of 8 to 12 animals). The preadipocyte cell line 3T3-L1 served as a positive and GAPDH as a loading control. (C) IB4 staining of WT-CPC after 4 weeks in culture (IB4, green; DAPI, blue; scale bar: 100 μm). (D) Oil Red O staining visualizes spontaneous differentiation of adipocytes in WT-CPC and CKO-CPC after 4 weeks of cultivation, scale bars: 50 μm. (E) Bar graph summarizing adipocyte counts of Oil Red O positive cells ($n = 5$ independent isolations; each isolation consists of 10 to 12 animals per genotype). (F–H) qRT-PCR detects mRNA levels of adipocyte markers (F) CEBPA, (G) FABP4, and (H) ZFP423 after 4 weeks of cultivation in WT-CPC and CKO-CPC ($n = 5$ (F and G) and $n = 3$ (H), independent cell isolations; each isolation consists of 10 to 12 animals per genotype). (I and J) qRT-PCR detects mRNA levels of WAT markers (I) LYZ2 and (J) resistin after in vitro cultivation in WT-CPC and CKO-CPC ($n = 3$ isolations per genotype). (K and L) qRT-PCR detects mRNA levels of BAT/BET markers (K) EBF2 and (L) TMEM26 after in vitro cultivation in WT-CPC and CKO-CPC ($n = 3$ isolations per genotype). (E–L) Bar graphs represent mean ± SD, mean of WT was set at 100%, * $p < 0.05$, ** $p < 0.01$ vs. WT, 1 sample $t$ test. Underlying data can be found in S1–S8 Data and S9 and S10 Figs. BAT, brown adipose tissue; BET, beige adipose tissue; CEBPA, CCAAT/enhancer-binding protein alpha; CKO, conditional knockout; CPC, cardiac progenitor cell; EBF2, early B cell factor 2; FABP4, fatty acid binding protein 4; GAPDH, glyceraldehyde 3-phosphate dehydrogenase; IB4, isolectin B4; PDGFRα, platelet-derived growth factor receptor alpha; PREF-1, preadipocyte factor 1; qRT-PCR, quantitative real-time PCR; TMEM26, transmembrane protein 26; WAT, white adipose tissue; WT, wild-type.

mice with normal cardiac function (S1 Table). A genome-wide exploratory methylation profiling by reduced representation bisulfite sequencing (RRBS) revealed a total of 83 differentially methylated regions (DMRs) overlapping with 81 unique genes with a minimum group methylation difference of 0.1 and $p < 0.001$ (S4A Fig, accession number PRJNA602737). DMRs with hypermethylated regions were detected in genes such as *Epas1* and *Fgfr2*, both known to promote angiogenesis [28, 29] (S4A Fig, Fig 3A and 3B). In turn, regions in the vicinity of the *Zfp423* 5-UTR (in exon 2, chr8: 87783293–87783352, Wilcoxon $p < 5.5 \times 10^{-6}$) were hypomethylated in CKO-CPC compared with WT-CPC (S4A Fig, Fig 3C). *Zfp423* expression is negatively regulated by zinc finger protein 521 (ZFP521) (30), which was slightly hypermethylated in CKO-CPC (Wilcoxon $p < 0.02$; S4A Fig, Fig 3D). Real-time quantitative PCR (qRT-PCR) confirmed higher mRNA expression of ZFP423 in CKO-CPC compared with that in WT-CPC, while no difference was observed for ZFP521 expression (Fig 3E and 3F). *Zfp423* expression can also be regulated by bone morphogenetic protein (BMP)2 and BMP4 [30, 31], but no differences were detected in the methylation pattern of these genes, and the mRNA expression of BMP2 and BMP4 in LV heart tissue taken from CKO and WT male mice was similar (Fig 3G and 3H). The mRNA levels of the EZH2 subunit of the PRC2, a histone methyltransferase associated with transcriptional repression of ZFP423 (25), was reduced in freshly isolated CKO-CPC compared with WT-CPC from male mice (Fig 3I). In turn, no alteration in EZH2 or ZFP423 mRNA levels as observed in CKO- and WT-CPC isolated from 3-month-old female mice (Fig 3J and 3K).

## Clonally expanded CPC have the potential to differentiate into endothelial cells and white adipocytes

As freshly isolated CPC are generally a heterogenic pool of cells [22], we tested whether the same CPC progenitor cell can differentiate into endothelial cells and adipocytes depending on the microenvironment. Using 2 clonally expanded CPC (cCPC) cell lines [22, 32], we observed that the same passage of cCPC cell lines differentiate into endothelial cells on Matrigel [22, 32] or into adipocytes when adipocyte differentiation media and the peroxisome proliferator activated receptor γ (PPARγ) activator indomethacin [33] were added (S5A and S5B Fig). Adipocyte differentiation led to a reduction in SCA-1 and PDGFRα expression and an up-regulation of the general adipocyte markers CEBPA and FABP4 (S5C–S5F Fig). Further analyses showed the up-regulation of the WAT markers LYZ2 and resistin, while the expression of the BAT/BET markers EBF2 and TMEM26 remained unchanged (S5G–S5J Fig), and PRDM16 and UCP-1 were undetectable.

## Retroviral overexpression of ZFP423 induced white adipocyte differentiation in cCPC

Retroviral overexpression of ZFP423 induced white adipocyte differentiation in cCPC as shown by increased Oil Red O staining and enhanced expression of PPARG2, CEBPA, LYZ2, and resistin (S6A–S6G Fig). The expression of the BAT/BET markers EBF2 and TMEM26 remained unchanged (S6H and S6I Fig), and PRDM16 and UCP-1 were not detectable.

## PGD$_2$ promotes white adipocyte differentiation of isolated WT-CPC

In order to analyze whether enhanced PGD$_2$ is responsible for increased white adipocyte formation in male CKO hearts, isolated WT-CPC were incubated with PGD$_2$, which induced white adipocyte differentiation (Fig 4A and 4B). This was associated with an early reduction of EZH2 and enhanced ZFP423 expression (Fig 4C and 4D). During cultivation, PGD$_2$ also

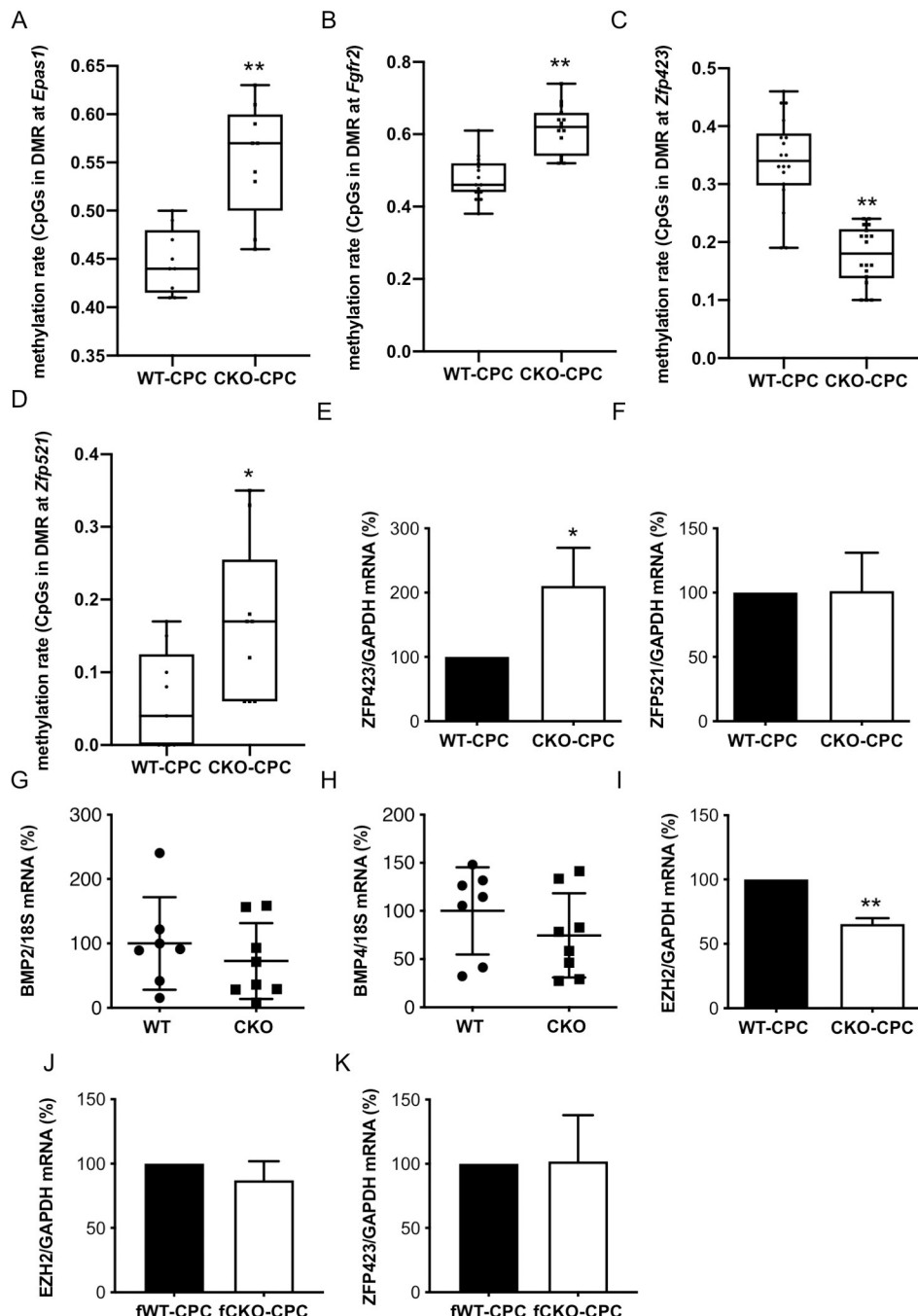

**Fig 3. Epigenetic analysis of freshly isolated CKO- and WT-CPC.** (**A** and **B**) Boxplots with single CpG methylation values for (**A**) *Epas1* (chr17:86759066–86759095) and (**B**) *Fgfr2* (chr7:**130728553**–130728597). (**C**) Boxplot with single CpG methylation values in a region overlapping with the 5′-UTR of exon 2 of *Zfp423* in CKO-CPC and WT-CPC (chr8: 87783293–87783352). Each group contains 3 samples with methylation values for 6 CpGs within the indicated genomic interval, respectively (*n* = 18). (**D**) Boxplot with single CpG methylation of a region approximately 1.5 kb downstream of the second exon of *Zfp521* in CKO-CPC and WT-CPC (chr18:13969640–13969667). For each of the 3 samples per group, 3 CpG methylation values in the identified region (*n* = 9) are shown. (**E** and **F**) Bar graphs summarize mRNA levels detected by qRT-PCR of (**E**) ZFP423 and (**F**) ZFP521 in freshly isolated WT-CPC and CKO-CPC (*n* = 5 independent isolations, each isolation consists of 8 to 12 animals). (**G** and **H**) Dot plots summarize (**G**) BMP2 and (**H**) BMP4 mRNA levels of WT (*n* = 7) and CKO (*n* = 8) LVs. (**I**) Bar graph summarizes EZH2 mRNA levels detected by qRT-PCR in freshly isolated WT-CPC and CKO-CPC from 3-month-old male (*n* = 5 independent isolations, each isolation consists of 8 to 12 animals). (**J** and **K**) Bar graphs summarize (**J**) EZH2 and (**K**) ZFP423 mRNA levels detected by qRT-PCR in freshly isolated CPC from 3-month-old female WT and CKO mice (CPC

isolated and pooled from 3 WT and 2 CKO mice). (**A–D**) Differences in methylation values were tested by Wilcoxon test, $^*$ $p < 0.05$, $^{**}$ $p < 0.001$. (**E–K**) Data are presented as mean ± SD, and the mean of WT was set to 100%, $^*$ $p < 0.05$ vs. control, $^{**}$ $p < 0.01$ vs. control; (**E, F, I–K**) 1 sample $t$ test and (**G and H**) 2-tailed unpaired $t$ test. Underlying data can be found in S1 Data. CKO, conditional knockout; CPC, cardiac progenitor cell; EZH2, enhancer of zeste homolog 2; qRT-PCR, quantitative real-time PCR; WT, wild-type.

enhanced expression levels of the adipocyte markers PPARG2, CEBPA, and FABP4 and of the WAT markers ZFP423, LYZ2, and resistin, while the expression of BAT/BET markers (EBF2 and TMEM26) remained unchanged (Fig 4E–4L), and PRDM16 and UCP-1 expression were not detectable.

## PGD$_2$-induced adipocyte differentiation in cCPC that could be prevented by the PGD$_2$ receptor 2 antagonist BAY-u 3405 but not by the PGD$_2$ receptor 1 antagonist BWA868C

The treatment of cCPC with PGD$_2$-induced adipocyte differentiation (Fig 5A and 5H). The PGD$_2$-induced adipocyte differentiation in cCPC could be attenuated by the PGD$_2$ receptor (DP)2 antagonist BAY-u 3405, while incubation with the DP1 receptor antagonist BWA868C did not influence PGD$_2$-induced ZFP423 expression or adipocyte differentiation (Fig 5A–5H).

## PGD$_2$ induced adipocyte differentiation from human iPSC

PGD$_2$ treatment for 48 h reduced EZH2 expression and increased ZNF423 expression in human induced pluripotent stem cells (iPSC) (S7A and S7B Fig). Prolonged treatment with PGD$_2$ induced also white adipocyte differentiation of iPSC (resistin and Oil Red O staining, S7C and S7D Fig). In addition, ZNF423 and CEBPA expressions were elevated, and EZH2 expression was reduced in PGD$_2$-treated human iPSC compared with control iPSC (S7F–S7H Fig). All aspects of adipocyte differentiation induced by PGD$_2$ in iPSC could be attenuated by BAY-u 3405 (S7E–S7H Fig).

## Treatment of young male CKO mice with the COX-inhibitor Ibuprofen or the DP2 receptor antagonist BAY-u 3405 reduced adipocyte differentiation of CKO-CPC

In order to evaluate whether PGD$_2$ is responsible for the enhanced adipocyte differentiation potential of CPC in CKO male mice, young male CKO mice before heart failure onset were treated either with the COX-inhibitor Ibuprofen or the DP2 antagonist BAY-u 3405 in vivo. After 2 weeks of treatment, PGD$_2$ levels were reduced in the supernatants of isolated CKO-CM of Ibuprofen but not BAY-u 3405-treated CKO mice compared with levels in the supernatants of CKO-CM isolated from control CKO mice (Fig 6A and 6B). CKO-CPC isolated from Ibuprofen- or BAY-u 3405-treated CKO mice displayed enhanced EZH2 expression and reduced ZFP423 expression compared with control CKO-CPC (Fig 6C–6F). The spontaneous adipocyte differentiation of CKO-CPC isolated from Ibuprofen or BAY-u 3405-treated CKO mice was abolished compared with CKO-CPC isolated from untreated CKO mice (Fig 6G and 6H).

## EPO reduces ZFP423 expression and adipocyte differentiation of CKO-CPC

We previously demonstrated that the addition of recombinant murine (rm)EPO to CKO-CPC cultures restored their endothelial differentiation potential (22). Here, we observed that

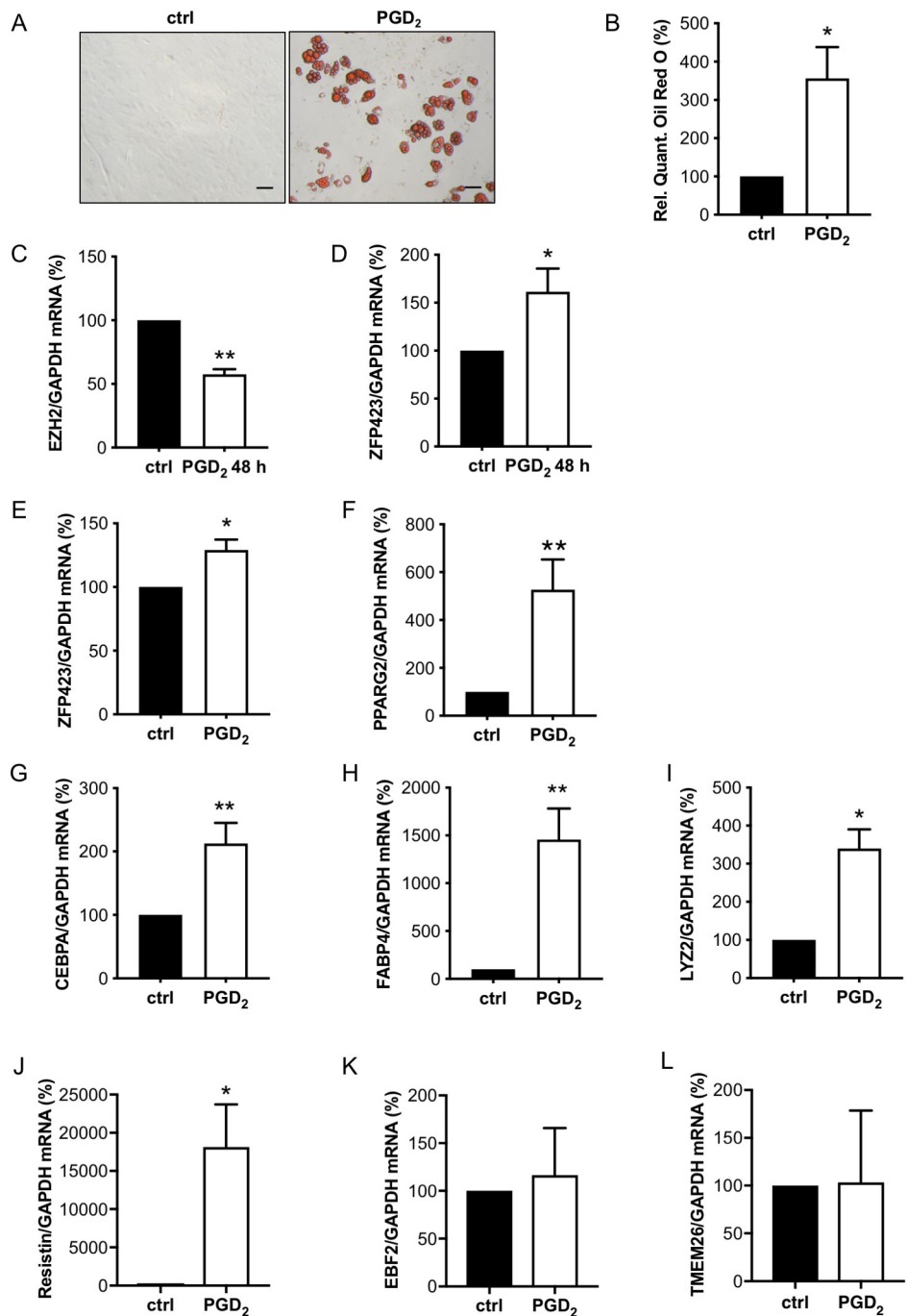

**Fig 4. PGD$_2$ promotes white adipocyte differentiation of male WT-CPC.** (**A**) Representative picture after Oil Red O staining of isolated WT-CPC treated with PGD$_2$ (1 μM) for 48 h and cultivated for 12 days, scale bars: 50 μm. (**B**) Relative quantification of Oil Red O measured by absorbance at 492 nm. (**C** and **D**) Bar graphs summarize mRNA levels assessed by qRT-PCR of (**C**) EZH2 and (**D**) ZFP423 in WT-CPC after treatment with PGD$_2$ for 48 h. (**E-H**) Bar graphs summarize mRNA levels assessed by qRT-PCR of (**E**) ZFP423, (**F**) PPARG2, (**G**) CEBPA, and (**H**) FABP4 in WT-CPC incubated with PGD$_2$ (1 μM) for 48 h and cultivated for 12 days. (**I–L**) Bar graphs summarize mRNA levels assessed by qRT-PCR of the white adipocyte markers (**I**) LYZ2 and (**J**) resistin and of the brown/beige adipocyte markers (**K**) EBF2 and (**L**) TMEM26 in WT-CPC incubated with PGD$_2$ (1 μM). (**C, D, I–L**) Data are presented as mean ± SD (WT-CPC isolated and pooled from 6 animals), and the mean of control cells was set to 100%, * $p < 0.05$, ** $p < 0.01$ vs. control, 1 sample $t$ test. (**B, E–H**) Data are presented as mean ± SD (WT-CPC isolated and pooled from 3 animals), mean of control cells was set to 100%, * $p < 0.05$, ** $p < 0.01$ vs. control, 1 sample $t$ test. Underlying data can be found in S1 Data. CEBPA, CCAAT/enhancer-binding protein alpha; CPC, cardiac progenitor cell; EZH2,

enhancer of zeste homolog 2; FABP4, fatty acid binding protein 4; LYZ2, lysozyme 2; $PGD_2$, prostaglandin $D_2$; PPARG2, peroxisome proliferator-activated receptor gamma isoform-2; qRT-PCR, quantitative real-time PCR; TMEM26, transmembrane protein 26; WT, wild-type.

addition of rmEPO to CKO cultures persistently reduced the ZFP423 expression in CKO-CPC (S8A and S8B Fig), which was associated with attenuated adipocyte differentiation, as shown by reduced Oil Red O positive cells and reduced expression of CEBPA and FABP4 (S8C–S8F Fig). However, the addition of rmEPO to isolated CPC for 48 h had no effect on EZH2 expression (S8G Fig).

## Discussion

CM-specific STAT3-deficiency in male but not female mice leads to age-related heart failure [14, 17]. Here, we provide evidence that this is caused in part by an impaired AR receptor signaling due to STAT3 deficiency, which leads to a higher secretion of $PGD_2$ by male but not female CKO-CM (Fig 7). $PGD_2$ subsequently induces an epigenetic shift of CPC differentiation from endothelial cells to white adipocytes, thereby promoting WAT deposits in hearts from aging male but not female CKO mice (Fig 7). The shift of CPC toward adipocyte differentiation could be attenuated by the treatment of CKO mice with the COX inhibitor Ibuprofen or the DP2 receptor antagonist BAY-u 3405, thereby confirming the role of $PGD_2$ for cardiac WAT generation in vivo. Since WAT generation is associated with more severe heart failure (6), these data shed light on sex-specific remodeling processes in the heart that may contribute to age-related heart failure in males.

$PGD_2$ is generated by multiple enzymatic steps from arachidonic acid, and COX-2 is a rate-limiting enzyme in this process [34]. COX-2 is elevated in CM from female and male CKO mice, an observation that contrasts with findings showing that STAT3 is a direct transcription factor of COX-2 in ischemic preconditioning, where the activation of cardiac STAT3 up-regulates COX-2 in the heart [35]. In fact, beneficial effects of COX-2 in protection of the heart from ischemic injury have been reported also indicating that the use of COX inhibitors may have adverse effects in the setting of ischemia/reperfusion (I/R) [36]. In contrast, protective effects of COX-2 inhibition on CMs have also been reported [37]. The present study analyzes pathomechanisms of age-related idiopathic or DCM for which pathomechanisms are different from myocardial infarction or I/R injury as, for example, canonical STAT3 signaling is not or only moderately activated. Our data suggest that inactive STAT3 acts either directly or indirectly as a negative regulator of COX-2, a feature that will be explored in future studies. However, PGs are regulated not only by COX enzymes but also by the PG-degrading enzyme HPGD [38], and here we observed the lower expression of HPGD in male CKO-CM but not in female CKO-CM. In this context, it has been shown that the AR can be activated by different ligands including interleukin (IL)-6 and testosterone [39]. The AR is expressed in CMs from various species including mouse and human [26,40,41], and STAT3 acts as a positive cofactor for AR signaling by directly interacting with amino acids 234 to 558 in the N-terminal domain of the AR [24]. The AR is expressed on HL-1 cells, and indeed, HPGD expression is lower in HL-1-STAT3-KD-CM compared with that in HL-1-ctrl-CM. Moreover, testosterone induced a marked increase in HPGD in HL-1-ctrl-CM, which was blunted in HL-1-STAT3-KD-CM. Thus, both enhanced COX-2 and sex-specific lowered HPGD production finally contribute to enhanced $PGD_2$ production in STAT3-deficient male CMs. This deregulated arachidonic metabolism results in enhanced $PGD_2$ production in CMs which could also be confirmed in vivo as the treatment of CKO male mice with the COX-inhibitor Ibuprofen lowering the $PGD_2$ secretion in CKO-CM. Thus, we provide evidence that enhanced COX-2 and sex-

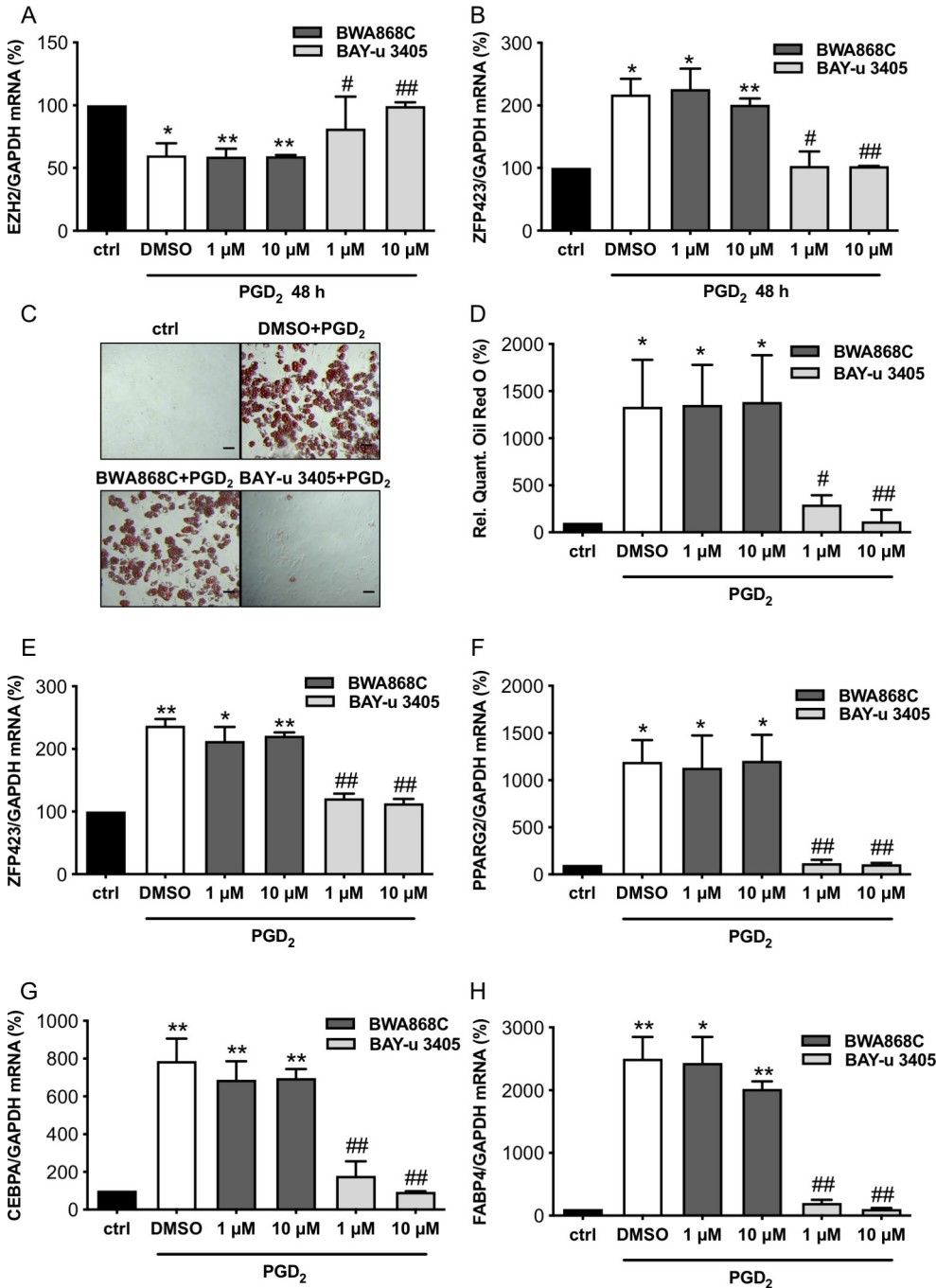

**Fig 5. PGD$_2$ induced adipocyte differentiation from cCPC is abrogated by DP2 antagonist BAY-u 3405 but not by the DP1 antagonist BWA868C.** Bar graphs summarizes (**A**) EZH2 mRNA expression and (**B**) ZFP423 mRNA expression of cCPC 48 h after treatment with the DP1 receptor antagonist BWA868C or the DP2 receptor antagonist BAY-u 3405 in indicated concentrations (1 μM and 10 μM) and PGD$_2$ stimulation (100 μM). (**C**) Representative pictures after Oil Red O staining of cCPC treated with DP receptor antagonists BWA868C (10 μM) or BAY-u 3405 (10 μM) and PGD$_2$ (100 μM) for 48 h and cultivated for 12 days, scale bars: 50 μm. (**D**) Relative quantification of Oil Red O measured by absorbance at 492 nm. (**E–H**) Bar graphs summarize mRNA levels assessed by qRT-PCR of (**E**) ZFP423, (**F**) PPARG2, (**G**) CEBPA and (**H**) FABP4 in cCPC incubated with DP receptor antagonists and PGD$_2$ (100 μM) for 48 h and cultivated for 12 days. Statistically significant differences between the groups are represented as mean ± SD, $n$ = 3 independent experiments, and the mean of mRNA expression levels of control cCPC were set at 100%, $^{**}$ $p < 0.01$ vs. ctrl, $^{*}$ $p < 0.05$ vs. ctrl, $^{##}$ $p < 0.01$ vs. PGD$_2$ and DMSO-treated cCPC, $^{#}$ $p < 0.05$ vs. PGD$_2$ and DMSO-treated cCPC, 1 sample $t$ test). Underlying data can be found in S1 Data. cCPC, clonally expanded CPC;

CEBPA, CCAAT/enhancer-binding protein alpha; CPC, cardiac progenitor cell; DP, PGD$_2$ receptor; EBF2, early B cell factor 2; PGD$_2$, prostaglandin D$_2$; PPARG2, peroxisome proliferator-activated receptor gamma isoform-2; qRT-PCR, quantitative real-time PCR.

specific reduced HPGD contribute to elevated PGD$_2$ production in male failure prone and failing hearts with low STAT3 expression (Fig 7).

A consequence of the enhanced cardiac PGD$_2$ secretion is an impairment of the endogenous cardiac regeneration potential as it shifts the differentiation potential of the CPC pool from endothelial cells toward white adipocytes. Indeed, isolated CKO-CPC display enhanced white adipocyte and reduced endothelial differentiation. Furthermore, depending on the differentiation conditions, cCPC can differentiate into endothelial cells or into adipocytes. These data indicate that not differences in the CPC populations but reduced STAT3 expression in CMs and subsequent impairment of the cardiac microenvironment, i.e., enhanced PGD$_2$ secretion and the previously reported reduced EPO secretion (22) promote the shift in their differentiation potential. The contribution of PGD$_2$ in this process is demonstrated by the observations that PGD$_2$ is able to induce white adipocyte differentiation in WT-CPC, in cCPC and in human iPSC. The white adipocyte formation by PGD$_2$ is mediated by the DP2 receptor since it could be inhibited by the DP2 receptor antagonist BAY-u 3405 but not by the DP1 receptor antagonist BWA868C. Moreover, since the treatment of CKO mice with the DP2 receptor antagonist BAY-u 3405 reduced the enhanced adipocyte formation of CKO-CPC, the role of PGD$_2$ via its DP2 receptor for the formation of WAT could also be confirmed in vivo.

The effect of increased PGD$_2$ on CPC appears to be longstanding, since isolated CKO-CPC from CKO male hearts maintain their increased white adipocyte differentiation capacity even if kept for several weeks under the same cultivation conditions as WT-CPC isolated from WT hearts.

Indeed, freshly isolated CKO-CPC or PGD$_2$-treated WT-CPC displayed an increased expression of the white adipocyte differentiation factor ZFP423. ZFP423 is a transcription factor controlling preadipocyte determination and maintenance of white adipocyte identity through suppression of the thermogenic gene program [8,10,12,42,43]. ZFP423 overexpression in cCPC was sufficient to induce white adipocyte differentiation confirming the important role of ZFP423 for the adipocyte commitment of CPC. Increased ZFP423 expression in male CKO-CPC was associated with an up-regulation of WAT markers (LYZ2 and resistin) and either unchanged expression (EBF2 and TMEM26) or no expression (PRDM16 and UCP-1) of BAT/BET markers. Finally, and in line with the sex-specific difference in PGD$_2$ production, CPC isolated from female CKO hearts did neither exhibit increased ZFP423 expression nor enhanced adipocyte differentiation.

BMP2, BMP4, and ZFP521 are known regulators of ZFP423, and ZFP521 suppresses the adipocyte lineage by direct transcriptional repression of ZFP423 [30,31]. In the present study, expressions of BMP2, BMP4, or ZFP521 were not altered in male CKO hearts or in freshly isolated male CKO-CPC compared with male WT hearts or WT-CPC, suggesting that these factors are not responsible for the ZFP423 regulation in male CKO-CPC. In turn, PGD$_2$ and its nonenzymatically generated metabolite 15-deoxy-delta(12,14)-prostaglandin-J$_2$ (PGJ$_2$) were reported to decrease EZH2 mRNA and protein expression [44]. EZH2 is a subunit of PRC2 and functions as a histone methyltransferase that is able to control CpG methylation through direct physical contact with DNA methyltransferases (DNMTs) [45]. Reduced EZH2 levels in the *Zfp423* promoter are associated with lower histone and DNA methylation and higher expression of the *Zfp423* gene, which predisposes fetal progenitor cells to adipogenic differentiation [46]. Indeed, in freshly isolated male CKO-CPC, the EZH2 expression was reduced and

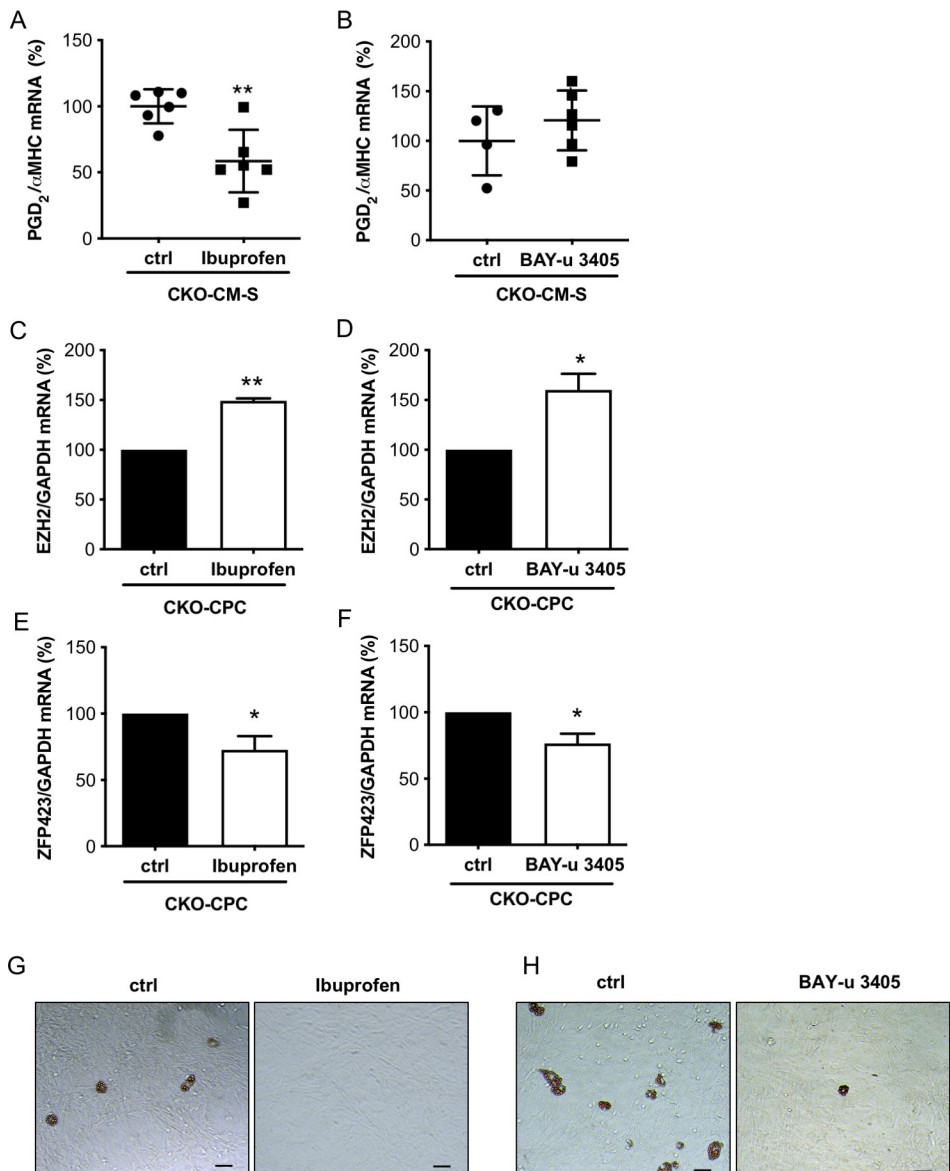

**Fig 6. Ibuprofen and BAY-u 3405 treatment of young male CKO mice enhanced EZH2 expression and reduced ZFP423 expression.** (**A** and **B**) Dot plots summarize $PGD_2$ levels in supernatants of CKO-CM isolated from young male CKO mice treated with Ibuprofen (**A**) or BAY-u 3405 (**B**) for 2 weeks and in supernatants of control CKO-CM. CKO-CM were isolated from 6 CKO mice treated with Ibuprofen and 6 respective control mice or 6 CKO mice treated with BAY-u 3405 and respective 4 control mice. (**C** and **D**) Bar graphs summarize mRNA expression of EZH2 of CKO-CPC isolated from (**C**) Ibuprofen-treated CKO mice and from control CKO mice or (**D**) BAY-u 3405-treated CKO mice and from control CKO mice. CKO-CPC isolated and pooled from 6 CKO mice treated with Ibuprofen and 6 respective control mice or 6 CKO mice treated with BAY-u 3405 and respective 4 control mice. (**E** and **F**) Bar graphs summarize mRNA expression of ZFP423 of CKO-CPC isolated from (**E**) Ibuprofen-treated CKO mice and from control CKO mice or (**F**) BAY-u 3405-treated CKO mice and from control CKO mice. CKO-CPC isolated and pooled from 6 CKO mice treated with Ibuprofen and 6 respective control mice or 6 CKO mice treated with BAY-u 3405 and respective 4 control mice. (**G** and **H**) Representative pictures after Oil Red O staining visualizing spontaneous differentiation of adipocytes in CKO-CPC after 4 weeks of cultivation. CKO-CPC isolated and pooled from 6 CKO mice treated with (**G**) Ibuprofen and 6 respective control mice or 6 CKO mice treated with (**H**) BAY-u 3405 and respective 4 control mice. scale bars: 50 μm. (**A**–**F**) Data are presented as mean ± SD, mean of CKO control was set to 100%, * $p < 0.05$ vs. control, ** $p < 0.01$ vs. control; (**A** and **B**) 2-tailed unpaired $t$ test and (**C**–**F**) 1 sample $t$test. Underlying data can be found in S1 Data. CKO, conditional knockout; CM, cardiomyocyte; EZH2, enhancer of zeste homolog 2; $PGD_2$, prostaglandin $D_2$.

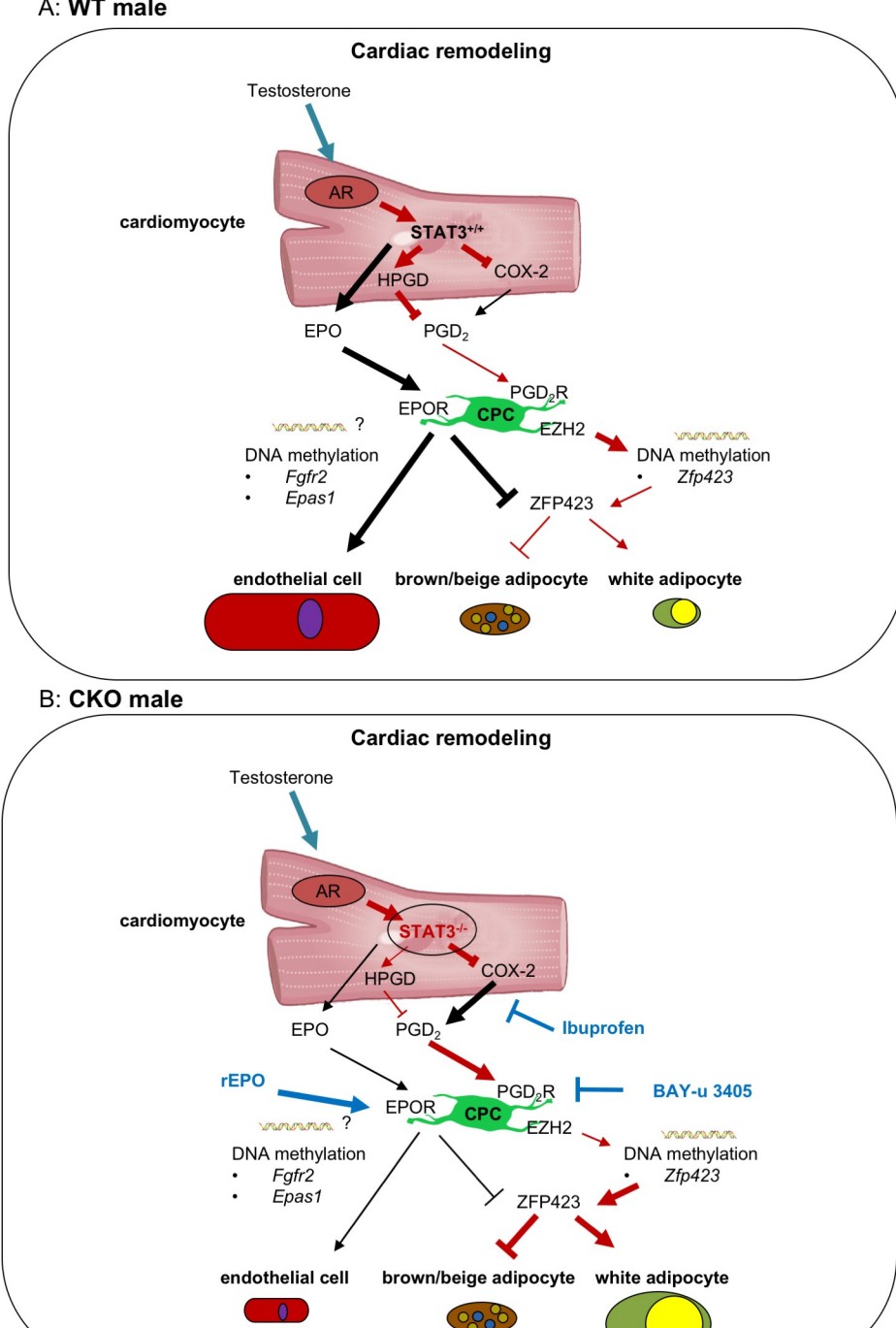

**Fig 7. Scheme explaining how CM STAT3-deficiency leads to enhanced white adipocyte differentiation of CPC in male mice contributing to sex-specific cardiac remodeling.** The present scheme shows in the upper picture (Fig 7A) the WT male and in the lower picture the CKO male cardiac signaling (Fig 7B). Sex-specific regulated pathways with dark red arrows and sex-unspecific regulated pathways with black arrows. CM-specific deficiency of STAT3 (Fig 7B) leads to enhanced expression of COX-2 in male and female mice whereas expression of HPGD is only reduced in male mice likely to result from impaired male-specific hormonal-mediated AR signaling caused by the missing AR cofactor STAT3 in male CKO-CM. This results in increased secretion of PGD$_2$ and of its metabolite PGJ$_2$ only in male CKO-CM. Subsequently, elevated PGD$_2$ levels in the cardiac microenvironment via activation of PGD2-RT2 reduced the expression of the EZH2 histone methyltransferase in CPC leading to reduced DNA methylation of its target gene *Zfp423* and with this to enhanced ZFP423 expression. ZFP423 promotes white and suppresses brown/beige adipocyte

differentiation of CPC in male CKO hearts. Reduced secretion of EPO from CKO-CM lowers the endothelial differentiation potential of CPC [22] which is associated with hypermethylation of the angiogenic genes *Epas1* and *Fgfr2*. In addition, to the previously shown beneficial effect of treatment of CKO mice with the EPO derivative CERA for endothelial differentiation (22), here we show that EPO also attenuates Zfp423 expression in CKO-CPC. Moreover, treatments of CKO mice with the COX inhibitor Ibuprofen or the PGD2-RT2 antagonist BAY-u 3405 suppress ZFP423 expression and adipocyte differentiation in CKO-CPC, thereby attenuating the endothelial-to-adipocyte shift of CPC. AR, androgen receptor; CKO, conditional knockout; CM, cardiomyocyte; CPC, cardiac progenitor cell; COX, cyclooxygenase; EPO, erythropoietin; EZH2, enhancer of zeste homolog 2; HPGD, hydroxyprostaglandin-dehydrogenase; PGD$_2$, prostaglandin D$_2$; STAT3, signal transducer and activator of transcription factor-3.

several DMRs were hypomethylated in the *ZFP423* gene compared with male WT-CPC. The role of PGD$_2$ for this epigenetic switch of CPC is supported by the observation that PGD$_2$ stimulation of isolated WT-CPC reduced EZH2 expression and increased ZFP423 expression. In addition, the treatment of young male CKO mice with the COX inhibitor Ibuprofen not only reduced the release of PGD$_2$ from CMs but also enhanced EZH2 and reduced ZFP423 expression and adipocyte differentiation in isolated CKO-CPC. Similarly, the treatment of CKO mice with the DP2 receptor antagonist BAY-u 3405 increased EZH2 and reduced ZFP423 expression and adipocyte differentiation in CKO-CPC further confirming the role of PGD$_2$ for the epigenetic switch of CPC toward adipocyte differentiation. Taken together, these observations extend previous studies reporting that PGD$_2$ suppresses lipolysis in adipocytes and is associated with insulin resistance and body weight gain [47]. This novel aspect of PGD$_2$ is interesting with regard to the controversial roles of PGD$_2$ documented for the cardiovascular system. In fact, it has been shown that PGD$_2$ not only has protective features for example in I/R injury [48] but may also have adverse effects, for example, by inducing CM death [49] and/or vasoconstriction [50].

Beside the hypomethylated *Zfp423* gene, male CKO-CPC displayed hypermethylated DMRs in genes promoting angiogenesis (for example, *Epas1* and *Fgfr2* [28, 29]) consistent with their previously shown lower endothelial differentiation potential [22]. Previous data showed that EPO secretion is reduced in the cardiac microenvironment from male CKO hearts and exogenous EPO restored the endothelial differentiation capacity in male CKO-CPC [22]. The present study found that EPO reduced ZFP423 expression and adipocyte formation from CKO-CPC, suggesting that lower EPO levels in CKO hearts may further promote the shift from endothelial to adipocyte differentiation in CKO-CPC. So far, there is no evidence for a sex-specific regulation of EPO since EPO blood levels in patients undergoing androgen deprivation therapy remained unchanged [51].

Previous data reported reduced STAT3 in failing human hearts [15], and the present study reveals significantly lower HPGD expression in LV tissue from male patients with terminal heart failure. Moreover, male patients but not female patients with heart failure due to DCM displayed elevated PGD$_2$ serum levels suggesting that sex-specific enhanced PGD$_2$ are also present in humans with heart failure. Furthermore, since PGD$_2$ induced ZFP423 expression and adipocyte differentiation in human iPSC, PGD$_2$ may also shift the differentiation potential of human tissue progenitor cells toward WAT formation.

In conclusion, age-related reduction of CM STAT3 leads to impaired AR receptor signaling and subsequent reduced expression of the PG-degrading enzyme HPGD (Fig 7). This results in a sex-specific increased secretion of PGD$_2$ from CMs in male but not female hearts (Fig 7). Subsequently, PGD$_2$ induces an epigenetic shift in CPC already prior to heart failure, which may hamper vascular regeneration and increase WAT deposits, thereby promoting adverse remodeling and heart failure (Fig 7). Finally, the present data suggest that the COX inhibitor Ibuprofen or the DP2 receptor antagonist BAY-u 3405 may attenuate these remodeling

process and, with this, reduce age-related heart failure specifically in males with impaired AR signaling.

Limitation to this study: Although, we could show that COX inhibition and more specifically inhibition of the $PGD_2$ receptor DP2 prevents the enhanced white adipocyte formation of CPC in failure prone CKO male hearts, we have not performed chronic treatment with COX inhibitors or BAY-u 3405 to test whether they would have an impact on age-related heart failure of the DCM phenotype.

## Material and methods

Unless otherwise stated, chemicals and reagents were all purchased from Sigma-Aldrich (Steinheim, Germany).

### Ethics statement—Humans

The local ethics committee of Hannover Medical School approved this study. All patients provided written informed consent. The study conforms to the principles outlined in the declaration of Helsinki.

The study was approved by the MHH local ethic committee (Nr. 1833–2013).

The generation of iPSC cells was approved by the MHH local ethic committee (Nr. 3242–2016).

### Ethic statement—Animals

All animal studies were conducted in accordance with the German animal protection law and with European Communities Council Directive 86/609/EEC and 2010/63/EU for the protection of animals used for experimental purposes. All experiments were approved by the Local Institutional Animal Care and Research Advisory Committee and permitted by the relevant local authority for animal protection "Niedersächsisches Landesamt für Verbrauerschutz und Lebensmittelsicherheit" (LAVES), application numbers 33.9-42502-04-08/1558,11/0552, 19/3328).

### Animal experiments

Mice with a CM-restricted knockout of STAT3 (CKO: αMHC-Cre[tg/+]; STAT3[flox/flox]) and WT mice (STAT3[flox/flox]) were generated as previously described [17]. Echocardiography was performed on 3- and 6-month-old lightly sedated mice (isoflurane inhalation 0.5%) using a Vevo 770 (VisualSonics, Toronto, Canada) as described (14).

### Ibuprofen and BAY-u 3405 treatment of CKO mice

Male CKO mice in the age of 9 to 10 weeks were treated with Ibuprofen (10 mg/kg bodyweight, Ibuprofen sodium salt, Sigma-Aldrich, dissolved in drinking water) or BAY-u 3405 (10 mg/kg bodyweight, Ramatroban, medchemexpress (Monmouth Junction, New Jersey, United States of America), dissolved in drinking water) for 2 weeks. CKO control animals received drinking water.

### Histology and immunostaining

For cardiac morphological analyses, hearts were embedded in Tissue Tek OCT and frozen at −80˚C. Interstitial collagen was analyzed in picro-sirius red F3BA-stained sections [17,52]. Inflammation was determined in LV cryosections with an antibody recognizing CD45 (BD Pharmingen 550539, BD Biosciences, San Jose, California, USA). Capillary density was

determined in transversely sectioned LVs by isolectin B4 (Vector, Burlingame, California, USA), counterstained with wheat germ agglutinin (WGA, Vector) and DAPI for nuclear stain [14,52]. For immunostainings using primary antibody recognizing perilipin (#9349, Cell Signaling Technologys, Danvers, Massachusetts, USA), resistin (ab119501, abcam, Cambridge, UK) or UCP-1 (ab10983, abcam, Cambridge, UK) cryosections were fixed in acetone, were washed 3 times with PBS, and blocked with 10% donkey serum and 0.3% Triton in PBS for 1 h at room temperature. Cryosections were stained with perilipin antibody (1:100), resistin antibody (1:50), or UCP-1 antibody (1:100) overnight at 4˚C. Cryosections were washed 3 times, and incubation with the secondary antibody Cy3-anti-rabbit (1,250, Jackson ImmunoResearch, Newmarket, Pennsylvania, USA) and counterstaining with WGA were performed for 2 h at room temperature. Nuclei were stained with DAPI Hoechst 33342 (Sigma-Aldrich, Steinheim, Germany). Images were acquired with AxioVert200M microscope, Axiovison software 4.8, Axio Observer 7, and Zen 2.6 pro software (Carl Zeiss, Jena, Germany) and with SP8 Leica Inverted confocal microscope.

## Isolation, characterization, and culture of SCA-1[+] cardiac progenitor cells

Isolation of SCA-1[+] cells from hearts of 3-month-old mice was performed as described previously [22] and is described in detail in S1 Text.

## DNA Isolation for RRBS

The DNA of freshly isolated CKO-CPC and WT-CPC of 3-month-old male mice was prepared using the DNeasy Blood and Tissue Kit according to the manufacturer's protocol (Qiagen, Venlo, the Netherlands).

## Reduced representation bisulfite sequencing (RRBS)

RRBS libraries for a total of $n$ = 6 samples (3 sample pools for CKO mice, 3 sample pools for WT mice, 10 to 12 mice per pool) were prepared using the Ovation RRBS Methyl-Seq System 1–16 with TrueMethyl oxBS (NuGen, Redwood City, California, USA), skipping the oxidation step but otherwise undertaken according to manufacturer's instructions. The average conversion rate across all samples was estimated to be 97%. Sequencing was performed on a NextSeq500 instrument at the Sequencing Core Facility of the Fritz-Lipmann-Institute Jena. The input DNA and concentration used for library preparation are both given in Table A in S1 Text.

## Data analysis of RRBS

**Preprocessing.** Sequencing yielded between 139M and 171M sequences (Table 1). After clipping using *cutadapt* [53] (version 1.18; parameters: quality cutoff, 20; overlap, 5: minimum length, 25; -u 7 -a AGATCGGAAGAGC), between 138M and 170M remained (Table 1). Before and after clipping, the quality was inspected using *fastqc* (v0.11.5).

**Read alignment and conversion rates.** Using the Bisulfite Analysis Toolkit (BAT; v.0.1) [54–56], reads were aligned with *segemehl* (v0.2.0) [54] to the Mus musculus reference genome GRChm38.p6 using standard parameters of the module "BAT_calling" in mode "-F 1." For the extraction of conversion rates, we used the module "BAT_calling" with *haarz* (v0.1.7) [54, 56] and *samtools* (v1.6). The filtering of resulting vcf files containing the conversion rates was done using the module "BAT_filter_vcf." On average, mapping rates were approximately 89% (Table A in S1 Text).

**Table 1. Read numbers and mapping statistics.**

| Sample | Raw reads | Clipped reads | mapping |
|---|---|---|---|
| Set1 CKO | 171274320 | 170175455 | 152837610 (89.81%) |
| Set1 WT | 147258040 | 146432432 | 131141572 (89.56%) |
| Set2 CKO | 143418782 | 142616983 | 127924806 (89.7%) |
| Set2 WT | 139188392 | 138358371 | 123566916 (89.31%) |
| Set3 CKO | 158225479 | 157255130 | 141428776 (89.94%) |
| Set3 WT | 157426725 | 156389740 | 140337521 (89.74%) |

CKO, conditional knockout; WT, wild-type.

**Calling of differentially methylated regions.** The methylation data obtained from the previous step was summarized using "BAT_summarize" after the adaptation of the source code to allow the processing of mice data (BAT_summarize_mouse). Specifically, the module was called by assigning all 3 WT samples to the control group and all 3 CKO samples to the case group (parameters—in1 Set1_WT_CPC.bedgraph, Set2_WT_CPC.bedgraph, Set3_WT_CPC.bedgraph—in2 Set1_CKO_CPC.bedgraph, Set2_CKO_CPC.bedgraph, Set3_CKO_CPC.bedgraph—groups control, case—h1 WT1, WT2, WT3—h2 CKO1, CKO2, CKO3). The module was run with *circos* [57] (v0.69–6) and bedGraphToBigWig (v4). The calling of differentially methylated regions was carried out with "BAT_DMRcalling" using *metilene* (v0.2–8; parameters -a control -b case -z "-m 3 -M 1000 -d 0.01 -v 0.0" -p 1 -d 0.01 -c 3) [58]. In order to increase the sensitivity in the given setup, the minimum amount of CpGs per DMR was reduced to 3, and the distance between CpGs within 1 DMR was increased to 1 kb. Subsequently, the raw *metilene* output was annotated with the extended ENSEMBL annotation for GRCm38.p6 (v.92) using *bedtools intersect* (v2.22.1) [59]. For the extension of the gene annotation, all annotated genes were extended at their 5′-end by 5 kb to include the promoter region.

A total of 5 DMRs with an false discovery rate (FDR)-adjusted *p*-value were reported. Due to the high homogeneity of the samples and the relatively small size, we decided to analyze 2 sets of regions filtered for 2 different unadjusted *p*-value criteria. The first set with a minimum group methylation difference of 0.1 and $p < 0.001$ yielded a total of 83 DMRs overlapping with 81 unique genes. To better inspect the CpG methylation within predicted DMRs ($n = 83$), we calculated the median methylation for the respective DMRs for each sample and visualized the data using R's pheatmap function (R version 3.4.1, pheatmap version 1.0.12) with standard parameters for clustering after centering and scaling in the row direction.

Sequencing data of the epigenetic analyses are available under accession number PRJNA602737 in the Sequence Read Archive. Due to the sample size and the exploratory nature of this exercise, we omitted the correction for multiple testing. Pooling the CpG methylation values for all 3 samples in the respective groups was used to confirm or reject methylation differences (Wilcoxon) in CKO-CPC compared with those in WT-CPC.

## SCA-1[+] cell cloning

SCA-1[+] cell cloning was described previously [22,32]. The protocol for clonal expansion of CPC is given in S1 Text.

## Retroviral vector-mediated expression of ZFP423

For retroviral vector-mediated expression of ZFP423 the pMSCVFLAG-ZFP423 was used [42]. pMSCVFLAG-ZFP423 was a gift from Bruce Spiegelman (Addgene plasmid # 24764;

RRID:Addgene_24764). The plasmid pMSCV-GFP was used for transduction of control cells. For generation of lentiviral supernatants, the ecotropic Phoenix Packaging cell line (kindly provided by G. Nolan, Stanford University, Stanford, California, USA) was used [60]. Briefly, $5 \times 10^6$ Phoenix packaging cells were seeded on 10-cm plates in Dulbecco's Modified Eagle Medium (DMEM) supplemented with 10% fetal bovine serum (FBS). On the following day, retroviral constructs (5 μg), together with constructs encoding pGagpol (M57) (10 μg) and pEnv K73 (2 μg), were transiently transfected into Phoenix cells by the calcium phosphate transfection method. Six hours after transfection, the media were replaced by DMEM supplemented with 10% FBS, and the cells were cultivated overnight. The media were removed and DMEM/F12 supplemented with 10% FBS was added and further incubated for 24 h. Supernatants were harvested and filtered through a 40-μm nylon filter and were either directly used for the transduction of CPC or stored at −80˚C. Viral supernatants were added to CPC clones for 6 h in the presence of Polybrene (4 μg/ml). Transduction efficiency based on GFP-expression was controlled in all experiments.

## Adipocyte differentiation of cCPC and isolated WT-CPC

For adipocyte differentiation, cCPC or isolated WT-CPC were seeded on 0.2% gelatin in DMEM/F12 with 10% fetal calf serum (FCS) supplemented with insulin-transferrin, sodium selenite (ITS) (10 μg/ml insulin, 5 μg/ml transferrin, and 5 ng/ml sodium selenite), recombinant human (rh) fibroblast growth factor (FGF) basic (10 ng/ml), and EGF (20 ng/ml). The medium was replaced every 2–3 days. Two days after confluence, adipocyte differentiation was induced in DMEM/F12 supplemented with 10% FBS, 1 μM dexamethasone (G-Biosciences, St. Louis, Montana, USA), 0.5 mM methylisobutylxanthine, 1 μg/ml insulin, and 1% penicillin-streptomycin (Thermo Fisher Scientific, Waltham, Massachusetts, USA). Indomethacin (100 μg/ml) or PGD$_2$ (1 μM) were added to the adipocyte differentiation medium. After 2 days, the adipocyte differentiation medium was removed, and cells were maintained in DMEM/F12 supplemented with 10% FBS and 1 μg/ml insulin. The cells were harvested in TRIzol (Thermo Fisher Scientific, Rockford, USA) for RNA isolation or were fixed with 4% paraformaldehyde for Oil Red O staining.

## Information on stimulation of HL-1 cells are provided in the supplemental methods

**Adipocyte differentiation of human iPSC.**   The protocols for the generation and for the adipocyte differentiation of human iPSC are given in S1 Text.

**Treatment of cCPC and human iPSC with DP receptor antagonists.**   cCPC and human iPSC were incubated with the DP1 receptor antagonist BWA868C and the DP2 receptor antagonist BAY-u 3405 (1 μM and 10 μM in DMSO, both from Cayman Chemicals, Ann Arbor, Michigan, USA) starting 2 h prior PGD$_2$ treatment. In addition, cells were treated with PGD$_2$ and DMSO. Cells were cultivated for 48 h or for 2 weeks.

**Endothelial differentiation of cCPC on Matrigel.**   The plates were coated with 50 μl/cm$^2$ Matrigel Basement Membrane Matrix (Corning, New York, USA) at 37˚C for 30 min. Clonal CPC (50,000 cells/cm$^2$) were cultivated in endothelial cell growth medium-2 (EGM-2) complete medium (Lonza, Basel, Switzerland) overnight. Cells were fixed using 4% paraformaldehyde.

**Isolation of murine adult primary cardiomyocytes and cell culture.**   Murine adult primary CMs were isolated and cultivated as described previously [61] from 3-month-old WT and CKO mice. Supernatants of adult primary CMs were collected after 48 h, centrifuged at

300$g$ for 10 min to deplete cell fragments and stored at −80˚C. Cells were harvested in TRIzol (Thermo Fisher Scientific) for RNA isolation.

**Isolation of RNA and qRT-PCR.** Total RNA was isolated with TRIzol (Thermo Fisher Scientific), and cDNA synthesis was performed as described previously [22]. qRT-PCR with SYBR green dye method (Brilliant SYBR Green Mastermix-Kit, Thermo Fisher Scientific) was performed with the AriaMx Real-Time PCR System (Agilent Technologies, Santa Clara, California, USA) as described (22). Expression of mRNA levels were normalized using the $2^{-\Delta\Delta CT}$ method relative to glyceraldehyde 3-phosphate dehydrogenase (GAPDH) or 18S. A list of qRT-PCR primers used in this study is provided in the supplemental methods (Tables B and C in S1 Text).

**Immunoblotting.** Immunoblots were performed according to standard procedures using SDS-PAGE [17] and the STAT3-antibody (#9139, Cell Signaling).

**Flow cytometry.** A total of $5 \times 10^5$ freshly isolated CPC were stained with PREF-1 antibody (AF8277, R&D Systems, Minneapolis, Minnesota, USA) and PDGFRα antibody (17-1401-81, eBioscience, San Diego, California, USA) for 15 min at room temperature. Flow cytometry was performed using the FACSCalibur (BD Biosciences).

**Immunocytochemistry.** Immunostainings using isolectin B4 (Vector Laboratories, California, USA) were performed according to standard procedures (22). For immunostainings using primary antibody recognizing perilipin (#9349, Cell Signaling) or resistin (ab119501, abcam, Cambridge, UK) cells were washed 3 times with PBS and blocked with 10% donkey serum and 0.3% Triton in PBS for 1 h at room temperature. Cells were stained with perilipin antibody (1:100) or resistin antibody (1:50) overnight at 4˚C. Cells were washed 3 times, and incubation with the secondary antibody Cy3-anti-rabbit (1:250, Jackson ImmunoResearch) was performed for 2 h at room temperature. Nuclei were stained with DAPI Hoechst 33342 (Sigma-Aldrich). Images were acquired with AxioVert200M microscope and Axiovison software 4.8, Axio Observer 7, and Zen 2.6 pro software (Carl Zeiss).

**Oil red O staining.** Oil Red O staining was performed as previously described [22]. A detailed description is provided in S1 Text.

**Adipogenesis detection assay.** Triglyceride accumulation in LV heart tissue was quantified by an adipogenesis detection assay (Abcam ab102513) according to the manufacturer's protocol.

**PGD$_2$ detection in supernatants of adult murine cardiomyocytes.** PGD$_2$ levels in the supernatants of primary adult murine CMs were measured using the prostaglandin D$_2$ ELISA kit (Cayman Chemicals, No. 512031) according to the manufacturer's protocols. PGD$_2$ levels in supernatants of primary adult murine CMs were normalized to αMHC mRNA levels of the cells.

**Patient data.** All patients (males, $N = 24$; females, $N = 15$) enrolled in the study were diagnosed with heart failure due to idiopathic DCM. The sex- and age-matched control collective consisted of healthy males ($N = 29$) and females ($N = 31$) who displayed normal left ventricular ejection fraction (LVEF) (>55%), blood pressure, and heart rates.

LV tissues were taken from patients undergoing heart transplantation due to end-stage heart failure caused by DCM or ischemic cardiomyopathy (ICM) (DCM/ICM, $n = 8$). LV tissue from donor hearts not suited for transplantation served as controls (non-failing [NF], $n = 6$).

**Blood tests.** Blood samples were collected in S-Monovette tubes containing ethylenediaminetetraacetic acid (EDTA, for plasma) or clot activator (for serum) in patients with the diagnosis for DCM. Plasma or serum were separated by centrifugation at 1500 rpm for 10 min, and aliquots were stored at −80˚C. Laboratory workup was performed as part of routine

analysis by hospital laboratories. $PGD_2$ serum levels were measured using the prostaglandin $D_2$ ELISA kit (Cayman Chemicals, No. 512031) according to the manufacturer's protocols.

**Statistical analyses.** Statistical analysis was performed using GraphPad Prism version 5.0a and 8.1.2 for Mac OS X (GraphPad Software, San Diego, California, USA). Normal distribution was tested using the D'Agostino normality test. Continuous data were expressed as mean ± SD. Comparison between 2 groups was performed using 1 sample *t* test or unpaired 2-tailed *t* test. When comparing more than 2 groups, we used Bonferroni's multiple comparison test after 1-/2-way ANOVA testing. A 2-tailed *p* value of <0.05 was considered statistically significant.

## Supporting information

**S1 Fig. Adipocyte formation in heart tissue of 3- and 6-month-old CKO mice.** (**A**) Capillary density (upper panel: IB4 (green)/WGA (red) and nuclear DAPI staining (blue), Sirius Red staining (middle panel) and CD45 positive infiltrates (lower panel: brown, co-stained with eosin) of LV cryosections of 6 m male WT or CKO mice, scale bars: 50 μm. (**B**) Capillary density determined as the ratio of capillaries to CMs in transversely sectioned male WT (*n* = 7) and CKO (*n* = 5) LVs. (**C**) Dot plot summarizes VE-cadherin mRNA levels, of male WT (*n* = 6) and CKO LVs (*n* = 5), mean of WT was set at 100%. (**D**) Quantification of fibrosis from WT (*n* = 6) and CKO (*n* = 5) LVs in arbitrary units (arb. units). (**E**) Dot plot summarizes COL1A1 mRNA levels, of male WT (*n* = 8) and CKO LVs (*n* = 7), mean of WT was set at 100%. (**F**) Dot plot summarizes ADGRE1 mRNA levels of male WT (*n* = 8) and CKO LVs (*n* = 7), mean of WT was set at 100%. (**G**) Immunofluorescence staining of perilipin (red), resistin (red), or UCP-1 (red) counterstained with WGA-FITC (green) and DAPI (blue) in cryosections of heart tissue (male 6 m CKO mice), scale: 25 μM. (**H**) Perilipin staining in LV cryosections of 3- and 6-month-old (m) WT or CKO female (f) mice, perilipin (red), WGA (green), and DAPI (blue): scale bars: 50 μm. (**B–F**) All data are mean ± SD, * *p* <0.05, ** *p* <0.01 vs. WT, 2-tailed unpaired *t* test. Underlying data can be found in S1 Data. CKO, conditional knockout; CM, cardiomyocyte; FITC, fluorescein isothiocyanate; IB4, isolectin B4; LV, left ventricular; UCP-1, uncoupling protein 1; VE, vascular endothelial; WGA, wheat germ agglutinin; WT, wild-type.
(TIF)

**S2 Fig. STAT3 deficiency alters COX-2 and HPGD expression in LVs of aged male CKO mice, whereas in females only COX-2 expression is altered.** (**A, E**) Dot plots summarize mRNA levels of (**A**) HPGD and (**E**) COX-2 in LVs of 6-month-old male WT (*n* = 8) and CKO mice (*n* = 7). (**B and F**) Dot plots summarize mRNA levels of (**B**) HPGD and (**F**) COX-2 in LVs of 3-month-old female WT (*n* = 7) and CKO mice (*n* = 4). (**C and G**) Dot plots summarize mRNA levels of (**C**) HPGD and (**G**) COX-2 in LVs of 6-month-old female WT (*n* = 6) and CKO mice (*n* = 6). (**D and H**) Dot plots summarize (**D**) HPGD and (**H**) COX-2 mRNA levels of isolated adult female WT-CM and CKO-CM (CM isolated and pooled from 3 WT and 2 CKO mice). (**A–H**) All data are mean ± SD, and WT mean was set at 100%, * *p* < 0.05, ** *p* < 0.01 vs. WT, 2-tailed unpaired *t* tests. Underlying data can be found in S1 Data. CKO, conditional knockout; CM, cardiomyocyte; COX, cyclooxygenase; HPGD, hydroxyprostaglandin-dehydrogenase; LV, left ventricular; STAT3, signal transducer and activator of transcription factor-3; WT, wild-type.
(TIF)

**S3 Fig. STAT3 deficiency alters COX-2 and HPGD expression in HL-1 CMs.** (**A**) Representative western blot showing protein expression of STAT3 in HL-1 control (ctrl) and

STAT3-KD cells. PS served as a loading control. (**B, C**) Bar graphs summarize mRNA levels assessed by qRT-PCR of (**B**) HPGD and (**C**) COX-2 in HL-1 cells treated with testosterone (10 nM) for 24 h. (**D, E**) Bar graphs summarize mRNA levels assessed by qRT-PCR of (**D**) HPGD and (**E**) COX-2 in HL-1 cells treated with estrogen (10 nM) for 24 h. (**B–E**) Data are presented as mean ± SD, $n = 4$, and the mean of HL-1-ctrl PBS was set at 100%, $^*$ $p < 0.05$, $^{**}$ $p < 0.01$ vs. HL-1-ctrl PBS, $^#$ $p < 0.05$, $^{##}$ $p < 0.01$ vs. HL-1-STAT3-KD PBS, 2-way ANOVA with Bonferroni's multiple comparison test. Underlying data can be found in S1 Data and S11 Fig. COX, cyclooxygenase; HPGD, hydroxyprostaglandin-dehydrogenase; KD, knockdown; PS, Ponceau S; qRT-PCR, quantitative real-time PCR; STAT3, signal transducer and activator of transcription factor-3.
(TIF)

**S4 Fig. Methylation analysis of freshly isolated CKO- and WT-CPC.** (**A**) Heatmap of median methylation values in predicted DMRs ($n = 83$) overlapping with 81 unique genes. Underlying data can be found in S1 Data. Sequencing data of the epigenetic analyses are available under accession number PRJNA602737 in the Sequence Read Archive. CKO, conditional knockout; CPC, cardiac progenitor cell; DMR, differentially methylated region; WT, wildtype.
(TIF)

**S5 Fig. Adipocyte differentiation and endothelial cell formation of cCPC expanded from single cells.** (**A, B**) Differentiation of cCPC expanded from single cells (**A**) on Matrigel (left panel: phase contrast, right panel: Isolectin B4 staining (green); scale bars indicate 100 μm) and (**B**) after adipogenic induction (upper left panel: phase contrast; upper right panel: perilipin staining [red], nuclear staining, DAPI [blue]; lower left panel: Oil Red O staining [red]; and lower right panel: perilipin staining [green], nuclear staining, DAPI [blue], Oil Red O staining [red]; scale bars indicate 50 μm). (**C–J**) mRNA levels of progenitor cell markers ((**C**) SCA1 and (**D**) PDGFRα), general adipocyte markers ((**E**) CEBPA, (**F**) FABP4), WAT markers ((**G**) LYZ2, (**H**) Resistin), and BAT/BET markers ((**I**) EBF2, (**J**) TMEM26) in undifferentiated and differentiated cCPC ($n = 5$ (**C–F**) and $n = 3$ (**G–J**) independent cell culture experiments. Statistically significant differences between the groups are represented as mean ± SD, and the mean of mRNA expression levels of undifferentiated cCPC were set at 100%, $^{**}$ $p < 0.01$ vs. control, $^*$ $p < 0.05$ vs. control, 1 sample $t$ test). Underlying data can be found in S1 Data. cCPC, clonally expanded CPC; CEBPA, CCAAT/enhancer-binding protein alpha; CPC, cardiac progenitor cell; EBF2, early B cell factor 2; FABP4, fatty acid binding protein 4; PDGFRα, platelet-derived growth factor receptor alpha; TMEM26, transmembrane protein 26; WAT, white adipose tissue.
(TIF)

**S6 Fig. Retroviral overexpression of ZFP423 leads to white adipocyte differentiation of cCPC.** (**A**) Oil Red O staining of cCPC with retrovirally mediated overexpression of ZFP423. Control cells were transduced with the pMSCV-GFP control virusplasmid, scale bars: 50 μm. (**B**) Relative quantification of Oil Red O measured by absorbance at 492 nm. (**C–E**) qRT-PCR detects mRNA levels of (**C**) ZFP423 and of the adipocyte markers (**D**) CEBPA and (**E**) PPARγ. (**F** and **G**) qRT-PCR visualizes mRNA levels of the white adipocyte markers (**F**) LYZ2 and (**G**) resistin. (**H** and **I**) qRT-PCR visualizes mRNA levels of the brown/beige adipocyte markers (**H**) EBF2 and (**I**) TMEM26. (**B–I**) Data are presented as mean ± SD, and the mean of pMSCV-GFP transduced control cells was set to 100% ($n = 6$ (**B–E**) and $n = 3$ (**F–I**) independent experiments), $^*$ $p < 0.05$, $^{**}$ $p < 0.01$ vs. pMSCV-GFP transduced control cells and 1 sample $t$ test. Underlying data can be found in S1 Data. cCPC, clonally expanded CPC; CEBPA,

CCAAT/enhancer-binding protein alpha; CPC, cardiac progenitor cell; EBF2, early B cell factor 2; LYZ2, lysozyme 2; PPARγ, peroxisome proliferator-activated receptor gamma isoform; qRT-PCR, quantitative real-time PCR; TMEM26, transmembrane protein 26;
(TIF)

**S7 Fig. Treatment of human iPSC with PGD$_2$ leads to white adipocyte differentiation.** (**A** and **B**), Bar graphs summarizes (**A**) EZH2 and (**B**) ZNF423 mRNA expression of human iPSC 48 h after PGD$_2$ stimulation (1 μM) and treatment with the DP2 receptor antagonist BAY-u 3405 in indicated concentrations (100 nm, 1 μM and 10 μM) for 48 h. (**C** and **D**) Representative pictures after (**C**) resistin (red) staining and (**D**) Oil Red O staining of PGD$_2$-treated human iPSC after 12 days; scale bars: 50 μm. (**E**) Bar graphs summarize the relative quantification of Oil Red O measured by absorbance at 492 nm. (**F–H**) Bar graphs summarize mRNA levels assessed by qRT-PCR of (**F**) EZH2, (**G**) ZNF423, and (**H**) CEBPA of PGD$_2$ (1 μM) treated-human iPSC incubated with BAY-u 3405 in indicated concentrations and cultivated for 12 days. $N = 3$ (**A, B, F–H**) and $n = 6$ (**E**) independent experiments. (**A, B, E–H**) Bar graphs represent mean ± SD, and the mean of control cells was set to 100%, * $p < 0.05$, ** $p < 0.01$ vs. ctrl, # $p < 0.05$, ## $p < 0.01$ vs. PGD$_2$ and DMSO-treated cells and 1 sample *t*test. Underlying data can be found in S1 Data. CEBPA, CCAAT/enhancer-binding protein alpha; DP, PGD$_2$ receptor; EZH2, enhancer of zeste homolog 2; iPSC, induced pluripotent stem cell; PGD$_2$, prostaglandin D$_2$; qRT-PCR, quantitative real-time PCR.
(TIF)

**S8 Fig. EPO supplementation prevents enhanced adipocyte formation in cultivated male CKO-CPC.** (**A, B**) Bar graphs summarize ZFP423 mRNA levels in isolated male CPC incubated with rmEPO (10 ng/ml) for (**A**) 48 h or (**B**) 4 weeks after isolation ((**A**): $n = 3$ cell isolations, each isolation consists of 8 to 12 mice per genotype and (**B**): $n = 5$ independent isolations, each isolation consists of 10 to 12 animals per genotype). (**C**) Oil Red O staining visualizes adipocytes in CKO-CPC cultures after 4 weeks of cultivation with or without the addition of rmEPO (10 ng/ml), scale bars: 50 μm. (**D**) Bar graph summarizing adipocyte counts ($n = 5$ independent isolations, each isolation consists of 10 to 12 animals per genotype). (**E, F**) qRT-PCR visualizes mRNA levels of the adipocyte markers (**E**) CEBPA and (**F**) FABP4 after 4 weeks of cultivation ($n = 3$ independent cell isolations, each isolation consists of 10 to 12 animals per genotype). (**G**) Bar graphs summarize EZH2 mRNA levels in isolated CPC incubated with rmEPO (10 ng/ml) for 48 h ($n = 3$ cell isolations, each isolation consists of 8 to 12 mice per genotype). (**A, B, D–G**) Data are presented as mean ± SD, and the mean of WT PBS was set at 100%, * $p < 0.05$, ** $p < 0.01$ vs. WT PBS, # $p < 0.05$, ## $p < 0.01$ vs. CKO PBS, 2-way ANOVA with Bonferroni's multiple comparison test. Underlying data can be found in S1 Data. CEBPA, CCAAT/enhancer-binding protein alpha; CKO, conditional knockout; CM, cardiomyocyte; CPC, cardiac progenitor cell; EPO, erythropoietin; EZH2, enhancer of zeste homolog 2; FABP4, fatty acid binding protein 4; qRT-PCR, quantitative real-time PCR; rmEPO, recombinant murine erythropoietin; WT, wild-type.
(TIF)

**S9 Fig. Figure showing the successive plots and gates that were applied to the FCS files (Fig 2A).**
(TIF)

**S10 Fig. The uncropped gel for Fig 2B.**
(TIFF)

**S11 Fig. The uncropped western blot for SFig 3A.**
(TIFF)

**S1 Data. Numerical raw data.** All numerical raw data are combined in a single excel file, "S1_Data.xlsx," This file consists of several spreadsheets and each spreadsheet contains the raw data of 1 figure.
(XLSX)

**S2 Data. FCS file for flow cytometry Sca-1 anti-rabbit APC (Fig 2A).**
(FCS)

**S3 Data. FCS file for flow cytometry Sca-1 CD140a (Fig 2A).**
(FCS)

**S4 Data. FCS file for flow cytometry Sca-1 IgG-APC (Fig 2A).**
(FCS)

**S5 Data. FCS file for flow cytometry Sca-1 (Fig 2A).**
(FCS)

**S6 Data. FCS file for flow cytometry Sca-1 Pref-1 APC (Fig 2A).**
(FCS)

**S7 Data. FCS file for flow cytometry AW (Fig 2A).**
(FCS)

**S8 Data. FCS file for flow cytometry Wash (Fig 2A).**
(FCS)

**S1 Table. Cardiac function and dimensions in male 3- and 6-month-old WT and CKO mice.** FS, LVEDD, LVESD, and heart rate (bpm) in 3- and 6-month-old male mice. Data expressed as mean ± SD, $^*$ $p < 0.05$, $^{**}$ $p < 0.01$ vs CKO 3 m, $^{##}$ $p < 0.01$ vs WT 6 m, 2-way ANOVA, Bonferroni's multiple comparison test. bpm, beats per minute; CKO, conditional knockout; FS, fractional shortening; LVEDD, left ventricular end-diastolic diameter; LVESD, left ventricular end-systolic diameter; WT, wild-type.
(DOCX)

**S2 Table. Cardiac function and dimensions in female 3- and 6-month-old WT and CKO mice.** FS, LVEDD, LVESD, and heart rate (bpm) in 3- and 6-month-old female mice. Data expressed as mean ± SD, n.s., 2-way ANOVA, Bonferroni's multiple comparison test. bpm, beats per minute; CKO, conditional knockout; FS, fractional shortening; LVEDD, left ventricular end-diastolic diameter; LVESD, left ventricular end-systolic diameter; WT, wild-type.
(DOCX)

**S3 Table. Summary of clinical data from DCM patients and healthy sex-matched controls.** NYHA, LVEF, BP, and NT-proBNP were analyzed in routine clinical lab tests. Gaussian distribution was tested by D'Agostino–Pearson omnibus normality test Comparison between the groups was performed using Student $t$ test for Gaussian distributed data (presented as mean ± SD) and the Mann Whitney U test where at least 1 column was not normally distributed (presented as median and range). BP, blood pressure; DCM, dilated cardiomyopathy; LVEF, left ventricular ejection fraction; NT-proBNP, N-terminal pro-brain natriuretic peptide; NYHA, New York Heart Association.
(DOCX)

**S1 Text. Supplemental Methods.**
(DOCX)

# Acknowledgments

We thank Martina Kasten, Silvia Gutzke, Birgit Brandt, Iris Dallmann, and Tim Kohrn for excellent technical assistance. We thank the research core unit for laser microscopy at the Hannover Medical School and the Core Facility of the Fritz-Lipmann-Institute Jena.

# Author Contributions

**Conceptualization:** Melanie Ricke-Hoch, Steve Hoffmann, Jean-Luc Balligand, Ofer Binah, Denise Hilfiker-Kleiner.

**Data curation:** Elisabeth Stelling, Melanie Ricke-Hoch, Maren Heimerl, Karin Battmer, Britta Stapel, Michaela Scherr.

**Formal analysis:** Elisabeth Stelling, Melanie Ricke-Hoch, Steve Hoffmann, Anke Katharina Bergmann.

**Funding acquisition:** Melanie Ricke-Hoch, Ofer Binah, Denise Hilfiker-Kleiner.

**Investigation:** Elisabeth Stelling, Maren Heimerl, Stefan Pietzsch, Britta Stapel, Michaela Scherr, Denise Hilfiker-Kleiner.

**Methodology:** Elisabeth Stelling, Melanie Ricke-Hoch, Sergej Erschow, Steve Hoffmann, Stefan Pietzsch, Karin Battmer, Alexandra Haase, Michaela Scherr, Jean-Luc Balligand.

**Resources:** Denise Hilfiker-Kleiner.

**Supervision:** Melanie Ricke-Hoch, Denise Hilfiker-Kleiner.

**Writing – original draft:** Elisabeth Stelling, Denise Hilfiker-Kleiner.

**Writing – review & editing:** Melanie Ricke-Hoch, Jean-Luc Balligand, Ofer Binah, Denise Hilfiker-Kleiner.

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
