## [Editor Report · Decision Letter 0]

20 Apr 2020

Dear Dr Hilfiker-Kleiner, 

Thank you for submitting your manuscript entitled "Increased prostaglandin-D2 in male but not female STAT3-deficient hearts shifts cardiac progenitor cells from endothelial to white adipocyte differentiation" for consideration as a Research Article by PLOS Biology.

Your manuscript has now been evaluated by the PLOS Biology editorial staff as well as by an academic editor with relevant expertise and I am writing to let you know that we would like to send your submission out for external peer review.

Please re-submit your manuscript within two working days, i.e. by Apr 22 2020 11:59PM.

Kind regards,

Di Jiang, PhD

Associate Editor

PLOS Biology

---

## [Decision Letter · Decision Letter 1]

27 May 2020

Dear Dr Hilfiker-Kleiner,

Thank you very much for submitting your manuscript "Increased prostaglandin-D2 in male but not female STAT3-deficient hearts shifts cardiac progenitor cells from endothelial to white adipocyte differentiation" for consideration as a Research Article at PLOS Biology. Your manuscript has been evaluated by the PLOS Biology editors, an Academic Editor with relevant expertise, and by two independent reviewers.

The reviews of your manuscript are appended below. You will see that the reviewers find the work potentially interesting. However, based on their specific comments and following discussion with the academic editor, I regret that we cannot accept the current version of the manuscript for publication. We remain interested in your study and we would like to consider resubmission of a comprehensively revised version that thoroughly addresses all the reviewers' comments. Importantly, we would like to see evidence that the PGD2-Zfp423 axis is a critical pathway driving adipogenesis in vivo (reviewer 2), and more in-depth analyses of the human tissue samples (reviewer 1, point 3). We cannot make any decision about publication until we have seen the revised manuscript and your response to the reviewers' comments. Your revised manuscript would be sent for further evaluation by the reviewers.

We appreciate that these requests represent a great deal of extra work, and we are willing to relax our standard revision time to allow you six months to revise your manuscript.We expect to receive your revised manuscript within 6 months.

**IMPORTANT - SUBMITTING YOUR REVISION**

*Resubmission Checklist*

*Published Peer Review*

*PLOS Data Policy*

*Blot and Gel Data Policy*

Sincerely,

Di Jiang, PhD

PLOS Biology

REVIEWS:

Reviewer #1: The study by Stelling et al. presents interesting results on how STAT3 deficiency drives left ventricular dysfunction during aging in male mice. The authors present clear evidence that a shift from differentiation of progenitor cells from endothelial cells towards adipocytes occurs in aged male STAT3 deficeint mice mainly driven by increased release of prostaglanin D2 due to increased activity of cyclooxygenase 2 and less activity of HPGD. The data are further supported by extensive in-vitro work on progenitor cells which most interestingly keep their phenotype even after long term cultivation. The experiments are well conducted and I have only a few points the authors might want to consider:

1. The release of PGD2 from isolated cardiomyocytes was normalized to total RNA. Why did the authors choose total RNA rather than a cardiomyocyte specific protein (troponin, alphaMHC) to account for cardiomyocyte number? Total RNA could still be derived from contaminating other cells. 

2. Did the authors verify the changes obeserved in the immunohistology (figure 1) on capillary density and fibrosis by other techniques (Western blot: CD31, collagen etc)? Both the changes in capillary density and fibrosis (which appears to be very small (1-3%)) are not very impressive and most likely does not account for heart failure development. 

3. The human data are interesting but very superficial. I am aware that obtaining human tissue might be complicated but at least the tissue available should be analyzed further: markers for endothelial cell content, adiopocyte content etc. Even more, some data from female heart failure patients would be of interest to proof that these parameters might differ depending on sex.

4. The authors demonstrate that COX2 might be a key protein in age-depnendent or post-partum triggered heart failure development in STAT3-deficiency. Does this mean that the phenotype can be rescued by COX2 inhibition? Did the authors try to treat aged male mice with COX2 inhibitors? In this context, the discussion on COX2 (PMID: 31734378; PMID: 31380437; PMID: 31278427 to name only a few recent studies) on ischemia-reperfusion related cardiac effects (which involves remodelling and heart failure development as well) needs to be extended.

Reviewer #2: Reduced expression and activity of cardiac STAT3 is linked to dilatative cardiomyopathy or peripartum cardiomyopathy; however, the mechanisms underlying pathologic cardiac remodeling triggered upon cardiomyocyte STAT3 inactivation has remained unclear. Here, the authors propose that cardiomyocyte-specific STAT3 deficiency leads increased PGD2

secretion from male cardiomyocyte but not from female cardiomyocytes. Increased PGD2 in the microenvironment then leads to suppression of EZH2 in cardiac progenitor cells (defined here as Sca-1+ positive cells) and subsequent activation of Zfp423, a transcription factor driving adipocyte differentiation. The authors propose that PGD2 driven adipogenesis from the Sca-1+ positive cells drive the ectopic adipocyte accumulation and loss of vascular density.

Overall, the manuscript is well written and the hypothesis is certainly novel and of general interest. The major concern lies in whether the states hypothesis is sufficiently supported by the data on hand. The authors demonstrate convincingly in vitro that PGD2 can activate Zfp423 expression in progenitor cells and promote adipogenesis. What is missing is evidence that this is the critical pathway driving adipogenesis in vivo in this model. In fact, any number of secreted factors from STAT3-deficient cardiomyocytes might influence adipogenesis. As such, the current observations establish associations, but not a clear sense of causality of the phenotype in vivo. For example, does inactivation of PGD2 receptor in Sca-1+ cells rescue the phenotype? Further evidence that this pathway is mediating the phenotype in vivo would strengthen the conclusions.

---

## [Decision Letter · Decision Letter 2]

4 Nov 2020

Dear Dr Hilfiker-Kleiner,

Thank you for submitting your revised Research Article entitled "Increased prostaglandin-D2 in male but not female STAT3-deficient hearts shifts cardiac progenitor cells from endothelial to white adipocyte differentiation" for publication in PLOS Biology. I have now obtained advice from the two original reviewers and have discussed their comments with the Academic Editor. 

We're delighted to let you know that we're now editorially satisfied with your manuscript. However before we can formally accept your paper and consider it "in press", we also need to ensure that your article conforms to our guidelines. A member of our team will be in touch shortly with a set of requests. As we can't proceed until these requirements are met, your swift response will help prevent delays to publication. Please also make sure to address the data and other policy-related requests noted at the end of this email.

- a cover letter that should detail your responses to any editorial requests, if applicable

*Copyediting*

*Published Peer Review History*

*Early Version*

Sincerely,

Ines

--

Ines Alvarez-Garcia, PhD,

Senior Editor,

ialvarez-garcia@plos.org,

PLOS Biology

ETHICS STATEMENT:

- Please include the full name of the IACUC/ethics committee that reviewed and approved the animal care and use protocol/permit/project license. Please also include an approval number.

Fig. 1B, D-L; Fig. 2A, E-L; Fig. 3A-K; Fig. 4B-L; Fig. 5A, B, D-H; Fig. 6A-F; Fig. S1B, C, D, E; Fig. S2A-H; Fig. S3B-E; Fig. S4; Fig. S5C-J; Fig. S6B-I; Fig. S7A, B, E-H; Fig. S8A, B, D-G

Thanks for depositing the sequencing data for epigenetic analyses in SRA - please make sure the data is made publicly available before the manuscript is accepted for production.

For figures containing flow cytometry data, we ask that you provide FCS files and a picture showing the successive plots and gates that were applied to the FCS files.

Reviewers’ comments

Rev. 1:

None.

Rev. 2:

The authors have made a good effort to address the concerns with new data. The manuscript is now much better suited for publication.

---

## [Editor Report · Decision Letter 3]

3 Dec 2020

Dear Dr Hilfiker-Kleiner,

On behalf of my colleagues and the Academic Editor, Cecilia W Lo, I am pleased to inform you that we will be delighted to publish your Research Article in PLOS Biology. 

PRODUCTION PROCESS

Before publication you will see the copyedited word document (within 5 business days) and a PDF proof shortly after that. The copyeditor will be in touch shortly before sending you the copyedited Word document. We will make some revisions at copyediting stage to conform to our general style, and for clarification. When you receive this version you should check and revise it very carefully, including figures, tables, references, and supporting information, because corrections at the next stage (proofs) will be strictly limited to (1) errors in author names or affiliations, (2) errors of scientific fact that would cause misunderstandings to readers, and (3) printer's (introduced) errors. Please return the copyedited file within 2 business days in order to ensure timely delivery of the PDF proof. 

If you are likely to be away when either this document or the proof is sent, please ensure we have contact information of a second person, as we will need you to respond quickly at each point. Given the disruptions resulting from the ongoing COVID-19 pandemic, there may be delays in the production process. We apologise in advance for any inconvenience caused and will do our best to minimize impact as far as possible.

EARLY VERSION

PRESS 

Kind regards,

Erin O'Loughlin

Publishing Editor, 

PLOS Biology

on behalf of

Ines Alvarez-Garcia,

Senior Editor

PLOS Biology